# Effects of intraspecific cooperative interactions in large ecosystems

A. Altieri [1,2*], G. Biroli[2]

**1** Laboratoire Matière et Systèmes Complexes (MSC), Université de Paris & CNRS, 75013 Paris, France
**2** Laboratoire de Physique de l'École normale supérieure, ENS, Université PSL, CNRS, Sorbonne Université, Université de Paris F-75005 Paris, France
* ada.altieri@u-paris.fr

## Abstract

We analyze the role of the *Allee effect* – a positive correlation between population density and mean individual fitness – for ecological communities formed by a large number of species. Our study is performed using the generalized Lotka-Volterra model with random interactions between species. We obtain the phase diagram and analyze the nature of the multiple equilibria phase. Remarkable differences emerge with respect to the logistic growth case, thus revealing the major role played by the functional response in determining aggregate behaviors of large ecosystems.

# 1   Introduction

In the last decades the field of theoretical ecology has gathered momentum fostered by an explosion of experimental results and increasingly sophisticated techniques [1–3]. More structured models have thus been proposed to integrate this plethora of empirical data [4] and to capture the main features of complex ecosystems. With respect to most pioneering studies carried out over the past decades, there is now a growing interest in systems composed by an enormous number of species that interact in myriad ways in very complex environments. Remarkably, in presence of a large number of interacting components such models can be rephrased through the prism of theoretical physics using concepts and methods rooted in statistical mechanics.

    One of the most recently studied framework in theoretical ecology is offered by the mean-field disordered version of the random Lotka-Volterra model describing the evolution of many randomly interacting species [5–12]. Despite its simple structure, such a model has been

recently proven to be able to reproduce critical behaviors and collective dynamics from many other ecological setups – which include notably cascade predation, plant pollinator, resource-consumer models [13] – as well as to be of great interest in interdisciplinary research domains such as genetics, epidemiology, evolutionary game theory [14, 15] up to the modelization of complex economies [16, 17]. For our purposes, the Lotka-Volterra equations offer a useful framework to study the so-called *Allee effect*, which is defined through a positive correlation between mean individual fitness (or per-capita growth rate) and population density over some finite interval as shown in Fig. 1 [18–20]. The Allee effect is called strong if there exists an initial population threshold below which population decreases, *i.e.* a species needs a sufficiently large initial population to avoid extinction. A weak Allee effect corresponds to the case in which no threshold exists but intraspecific cooperativity leads to an initial increase of the growth rate when population increases, see Fig. 1.

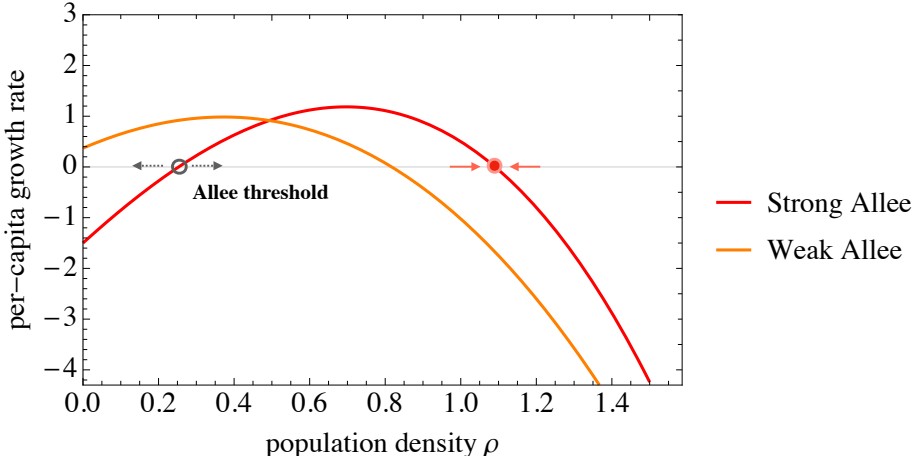

Figure 1: Pictorial representation of a *strong Allee effect* (in red) associated with a population decline at small population density below a finite threshold. The red dot represents a stable fixed point coinciding with the single species carrying capacity whereas the black empty dot is an unstable fixed point. In the case of a weak Allee effect (in light orange), no threshold exists leading to a definitely positive growth rate even at zero density.

The Allee effect, which inherited the name from the famous zoologist Allee, is based on the observation that in many species undercrowding, and not only competition, contributes to limiting the population growth. The behavioural observations, together with the original data on water loss and oxygen consumption [21], are used to interpret the curvilinear growth rate response to density and also to motivate the fact that aggregation can thrive the dynamics and have positive effects on the survival of a given ecosystem, as originally observed for land isopods that, if isolated, experience rapid desiccation. Allee aimed specifically at ascertaining the factors responsible for formation and evolution of animal aggregations, which in turn can contribute to increasing individual survival[1]. Caused by exploitation difficulties, social dysfunctions, predator saturation, genetic disease, this effect has been empirically confirmed in several aquatic, reptilian and mammalian populations [22, 23]. Furthermore, its undoubted

---

[1]Originally Allee thought that animals were unconscious of such benefits, for this reason he always preferred the expression *proto-cooperation* instead of *cooperation*.

centrality of is also linked to a better understanding of mechanisms governing the development of epidemic diseases [24][2] and cancer cell evolution [25]. Indeed, although the adaptive phenotypic switching between cell migration and proliferation plays a crucial role in tumors, it remains unclear how a particular phenotypic plasticity can affect overall tumor growth.

In the case of microbial ecosystems, species not only compete for common resources but may also display mutualism as a consequence of metabolic cross-feeding[3]. Such mutualistic interactions can give rise to a bistable behavior, *i.e.* two stable fixed points for the population growth, which turns out to be strictly connected to the existence of an initial growth rate threshold as a function of the population size. The Allee effect plays also an important role in spatially extended ecosystems – as it has been observed recently in a system of two cross-feeding species whose mutualistic strength is modulated by the inflow of nutrients [27]. When spatial fluctuations are taken into account, the Allee effect can counteract the genetic drift of a species and give rise to a pushed wave rather than a pulled wave, which instead would emerge from a simple logistic growth model.

From a more general perspective, our study addresses the question of the role of the functional response in the behavior of ecosystems formed by a large number of species. Basic models in population ecology rely on the logistic growth hypothesis. Deterministically, in the standard logistic model a small population grows exponentially up to saturation to a steady state value corresponding to its carrying capacity. But what if the form of the functional response is different? What are the main changes in the properties of the equilibria and of the complex dynamics displayed by the ecosystem for more general functional responses? Our study provides a first study of this general problem.

By using techniques rooted in statistical physics, and in particular in the theory of disordered systems, we provide a complete analysis of the generalized Lotka Volterra model for ecological communities formed by species with an Allee functional response. Our results, obtained for symmetric interactions and finite demographic noise in the species pool, open the route to a systematic understanding of the role of the functional response in determining collective movements, chaotic dynamics as well as generic system's instability of large ecosystems.

## 2   The model

In the following we define the model we focus on in this work, which consists in the generalized Lotka-Volterra equations for ecological communities. We then show a mapping to an equilibrium statistical physics problem, which is a key ingredient in our approach.

### 2.1   Disordered Generalized Lotka-Volterra Equations

We shall investigate the Allee effect by the introduction of a cubic one-species potential $V_i(N_i)$, which essentially modifies the logistic trend in favour of a multiplicative Allee effect [28]. We consider the following dynamical equations for the evolution of the relative species abundance $N_i$ at time $t$, where the index $i$ runs over the total number of species $i = 1, ..., S$ in the species

---

[2]Analyzing the relationship between the spread of an infection and a strong Allee effect may indeed be crucial to predict a catastrophic collapse of the endemic equilibria of a population.

[3]It has recently been shown that mutual cross-feeding plays a two-fold role: preventing competitive exclusion process but also reducing the energetic cost through the possibility of sharing efforts in amino acid synthesis [26].

pool:

$$
\frac{dN_i}{dt} = N_i \left\{ \rho_i \left[ -3N_i^2 + 2N_i(K_i + m) - K_i m \right] - \sum_{j,(j \neq i)} \alpha_{ij} N_j \right\} + \eta_i(t) + \lambda_i =
$$

$$
\frac{dN_i}{dt} = N_i \left[ -\nabla_{N_i} V_i(N_i) - \sum_{j,(j \neq i)} \alpha_{ij} N_j \right] + \eta_i(t) + \lambda_i \ ,
\tag{1}
$$

where $\rho_i \equiv r_i/K_i$ denotes the ratio between the single-species growth rate and the carrying capacity, and we follow Ito's convention for the stochastic equation. The additional parameter $\lambda$ denotes the immigration that at a first level analysis we assume to be species-independent, and $\eta_i(t)$ is a white noise with zero mean and covariance $\langle \eta_i(t)\eta_j(t') \rangle = 2TN_i(t)\delta_{ij}\delta(t-t')$, $T$ being the amplitude of the noise. $N_i(t)$ is interpreted as a relative species abundance of species $i$ at time $t$, meaning that the population is actually normalized by the total number of individuals populating the ecosystem in the absence of any interaction ($\alpha_{ij} = 0$). Therefore, since the amplitude $T$ turns out to be inversely proportional to the total number of individuals, *i.e.* $T = 1/N_{\text{ind}}$, one can tune such a control parameter to properly describe the demographic noise in a continuous setting [29, 30]. The larger the number of individuals the smaller the amplitude of demographic noise.

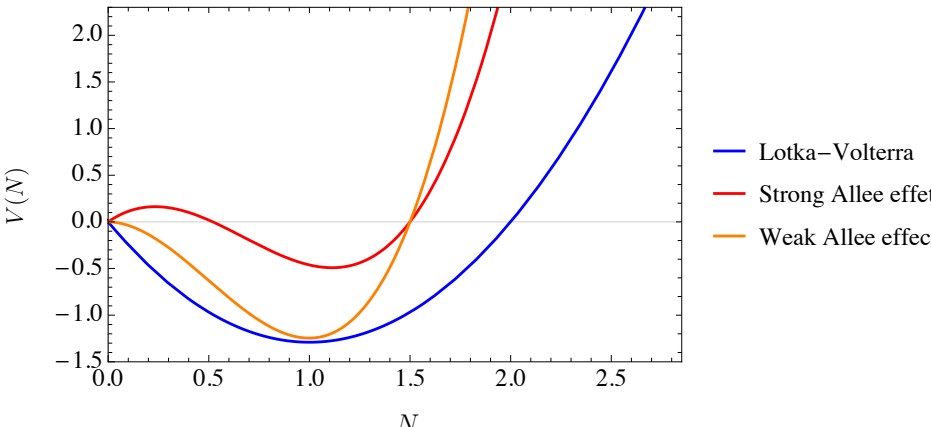

Figure 2: Plot of the Lotka-Volterra one-species potential $V(N) = -\rho(KN - \frac{N^2}{2})$ in blue, and of the modified cubic potential in Eq. (2) in orange and red corresponding to the growth rates shown in Fig. 1.

We model the forcing term in Eq. (1) through a cubic one-species potential

$$
V_i(N_i) = -r_i \left( 1 - \frac{N_i}{K_i} \right) (N_i - m) N_i
\tag{2}
$$

where the threshold value $m$ can be properly tuned to move from a strong ($m > 0$) to a weak ($-K_i \leq m \leq 0$) Allee effect. In Fig. (2) we compare the effect of a cubic potential with the standard Lotka-Volterra model corresponding to a quadratic dependence in $N_i$. According to this choice of the potential, the model admits three fixed points, respectively: i) at $N_i^* = 0$,

corresponding to the extinction, ii) at the threshold $m$; ii) finally, at the carrying capacity $N_i^* = K_i$. If the parameter $m$ is forced to be positive, the per-capita growth rate function displays a negative trend for $0 < N_i < m$ and a positive, increasing behavior up to the other (stable) fixed point in $K_i$.

Coming back to Eq. (1), the first term on the r.h.s. is then responsible for a self-regulating mechanism for the species abundances. The second term in parenthesis embeds the contribution due to the interactions among species. Following [6, 31], the elements of the matrix $\alpha_{ij}$ are i.i.d random variables distributed according to a Gaussian law with mean and variance respectively:

$$\text{mean}[\alpha_{ij}] = \mu/S \qquad \text{var}[\alpha_{ij}] = \sigma^2/S \; , \tag{3}$$

with $\alpha_{ij} = \alpha_{ji}$. The phases displayed by this model in the case of a logistic growth rate were studied in [6, 7, 11]. Here we instead focus on the Allee growth rate. Although we consider symmetric interactions, some of our results can be extended with a small degree of asymmetry, as we shall discuss in the Conclusions.

## 2.2 Dynamics-statics mapping via the Fokker-Planck formalism

Given the dynamics (1), we aim to write the corresponding Fokker-Planck equation for a generic observable $\mathcal{A}$ and to show that in the case of a symmetric interaction matrix $\alpha_{ij}$ it admits an invariant equilibrium-like probability distribution, as it was shown in [7]. We briefly recap the main steps here to make the interested reader aware of the procedure to obtain the corresponding Hamiltonian operator.

By using Ito's rule, the time derivative of a generic observable $\mathcal{A}$ can be written as:

$$\frac{d}{dt}\langle \mathcal{A}(\{N_j\})\rangle = \langle \sum_i \frac{\partial \mathcal{A}}{\partial N_i}\frac{dN_i}{dt}\rangle + T\langle \sum_i \frac{\partial^2 \mathcal{A}}{\partial N_i^2}N_i^2\rangle \; , \tag{4}$$

where $\langle \cdot \rangle$ stands for the average over the probability distribution $P(\{N_j\}, t)$, hence over the thermal noise. We can plug Eq. (1) in the above expression and obtain:

$$\frac{d}{dt}\langle \mathcal{A}(\{N_j\})\rangle = \left\langle \sum_i \frac{\partial \mathcal{A}}{\partial N_i}\left[-N_i\nabla_{N_i}V_i(N_i) - N_i\sum_j \alpha_{ij}N_j + \lambda\right]\right\rangle + T\langle \sum_i \frac{\partial^2 \mathcal{A}}{\partial N_i^2}N_i\rangle \; . \tag{5}$$

For the sake of compactness, we redefine the three terms in parenthesis as $\tilde{F}(\{N_j\})$, so that the equation for the evolution of the average operator $\mathcal{A}$ becomes

$$\frac{d}{dt}\langle \mathcal{A}(\{N_j\})\rangle = \int \prod_i dN_i P(\{N_j\}, t)\left[\sum_i \frac{\partial \mathcal{A}}{\partial N_i}\tilde{F}(\{N_j\}) + T\sum_i \frac{\partial^2 \mathcal{A}}{\partial N_i^2}N_i\right] \; , \tag{6}$$

from which the resulting equation for the evolution of $\langle \mathcal{A}(\{N_j\})\rangle$ can be easily obtained just by integrating by parts. In the same spirit, the dynamical equation for the probability distribution reads

$$\frac{\partial P(\{N_j\}, t)}{\partial t} = \sum_i \left[-\frac{\partial}{\partial N_i}\left[\tilde{F}(\{N_j\})P(\{N_j\}, t)\right] + T\frac{\partial^2}{\partial N_i^2}\left[P(\{N_j\}, t)N_i\right]\right] \tag{7}$$

from which we only need to impose the l.h.s to be zero in order to obtain the equilibrium probability. Therefore, we ask the invariant probability distribution to be $P = Z^{-1}\exp(-\beta H)$, $H$

being the associated energy function and $\beta = 1/T$ the inverse temperature. The condition for obtaining the quasi-stationary probability distribution implies $\frac{1}{P}\frac{\partial P}{\partial N_i} = -\frac{1}{T}\frac{\partial H}{\partial N_i}$ from which, by integrating over $N_i$, we can recover the expression for the Hamiltonian valid in the symmetric case:

$$H = \sum_i V_i(N_i) + \sum_{i<j} \alpha_{ij} N_i N_j + (T - \lambda) \sum_i \ln N_i \ . \tag{8}$$

Therefore, the original dynamical process in Eq. (1) defines the time evolution of an ecosystem with many interacting species whose thermodynamics is described by the Hamiltonian (8). Note that at variance with the simplest scenario corresponding to the analysis in the limit $T \to 0$ and $\lambda \to 0^+$, the introduction of a small but finite immigration results in a qualitative behavioral change. The immigration parameter ensures indeed that no species will go completely extinct from the ecosystem replacing the concept of indefinite extinction with a probabilistic argument, *i.e.* with the probability that at a given time a species is present or absent.

On a mathematical level, the parameter $\lambda$ provides a regularization of the probability distribution at small $N_i$ once the demographic noise is also taken into account. In order to see how this comes about, let's set $\alpha_{ij} = 0$ and specialize our reasoning to the single-species case. Taking in mind Eq. (2), the stationary distribution turns out to be $P_\infty(N) \propto N^{\lambda/T-1} \exp\left[\frac{1}{T}\left(-mN + N^2(1+m) - N^3\right)\right]$, according to which the effective potential can be written as $V_{\text{eff}}(N) = -mN - N^2(1+m) + N^3 - (\lambda - T)\ln N$. In the following we shall consider the case of a small but finite immigration rate, so that a stable steady state exists in presence of demographic noise. As done in [11], we impose a hard-core repulsive boundary corresponding to an infinite potential at $N_c = 10^{-2}$, which is numerically equivalent to having an immigration rate $\lambda = 10^{-2}$ but simpler to handle analytically.

# 3 Thermodynamic analysis with demographic noise: introduction to the replica method

As we have shown, the steady state properties of the ecosystem we are considering correspond to the thermodynamics equilibrium properties of a disordered system with the Hamiltonian (8). We can then take advantage of the replica method [32,33] developed in statistical physics of disordered system to carry out our study. The introduction of replicas allows us to derive the replicated free energy by the identity

$$-\beta F = \lim_{n \to 0} \frac{\ln \overline{Z^n}}{n} \ . \tag{9}$$

Operatively, one should compute the quantity on the r.h.s for integer values of the replica number $n$, then consider the analytical continuation to real values and eventually take the limit $n \to 0$. The starting point is the the computation of the replicated partition function:

$$\overline{Z^n} = \int \prod_{i,(ij)} dN_i^a \, d\alpha_{ij} \exp\left[-\sum_{(ij)} \frac{(\alpha_{ij} - \mu/S)^2}{2\sigma^2/S} - \beta H(\{N_i^a\})\right] \tag{10}$$

where the average over the disorder corresponds to computing the integral over the $\alpha_{ij}$s. By following usual steps in the physics of disordered systems, see *e.g.* [7,11] for very similar computations, one finds that the free energy can be written as an integral over order parameters:

$$F = -\frac{1}{\beta n} \ln \int \prod_{a,(a<b)} dQ_{ab} dQ_{aa} dH_a \; e^{S\mathcal{A}(Q_{ab}, Q_{aa}, H_a)} \; . \tag{11}$$

where the overlap matrix $Q_{ab}$ (with diagonal value $Q_{aa}$) and the external field $H_a$ are defined as:

$$Q_{ab} = \frac{1}{S} \sum_{i=1}^{S} N_i^a N_i^b \; , \tag{12}$$

$$H_a = \frac{1}{S} \sum_{i=1}^{S} N_i^a \; , \tag{13}$$

with $(a, b) = 1, ..., n$. Note that, at the variance with the partition function, the free energy becomes a self-averaging quantity in the thermodynamic limit, as $S \to \infty$, loosing any dependence on the specific realization of the disorder. Self-averaging holds for the majority of the properties of the system.

## 3.1 Effective Hamiltonian and Replica symmetric ansatz

The resulting action in Eq. (11) reads

$$\mathcal{A}(Q_{ab}, Q_{aa}, H_a) = -\rho^2 \sigma^2 \beta^2 \sum_{a<b} \frac{Q_{ab}^2}{2} - \rho^2 \sigma^2 \beta^2 \sum_a \frac{Q_{aa}^2}{4} + \rho \mu \beta \sum_a \frac{H_a^2}{2} + \frac{1}{S} \sum_i \ln Z_i \; , \tag{14}$$

where the last piece can be written as a Boltzmann measure in terms of an effective Hamiltonian

$$Z_i = \int \prod_a dN_i^a \exp\left(-\beta H_{\text{eff}}(\{N^a\}_i)\right) \; . \tag{15}$$

The effective Hamiltonian $H_{\text{eff}}$ corresponds to

$$\begin{aligned}
H_{\text{eff}}(\{N^a\}_i) = & -\beta \rho^2 \sigma^2 \sum_{a<b} N_i^a N_i^b Q_{ab} - \beta \rho^2 \sigma^2 \sum_a (N_i^a)^2 \frac{Q_{aa}}{2} + \\
& + \sum_a \left[\rho \mu H_a N_i^a + V_i(N_i^a) + (T - \lambda) \ln N_i^a\right] \; ,
\end{aligned} \tag{16}$$

which can be further simplified using some specific approximations.

The simplest scenario in the panorama of all possible replica techniques corresponds to the replica symmetric (RS) computation, which holds as long as the free-energy landscape is characterized by one single equilibrium. The overlap matrix is thus parametrized by two values: the self-overlap between replicas inside the same state, $q_d$, and the inter-state overlap, $q_0$. As for the external field, it is assumed to be uniform $\forall a$. This Ansatz translates into the property that any permutation of the replica indices does not affect the overall matrix structure.

$$\begin{aligned}
Q_{ab} &= q_0 \quad && \text{if} \quad a \neq b \\
Q_{aa} &= q_d \quad && \text{if} \quad a = b \\
H_a &= h \quad && \forall a
\end{aligned} \tag{17}$$

Accordingly, the action $\mathcal{A}$ in Eq. (14) becomes:

$$\mathcal{A}(q_d, q_0, h) = -\rho^2 \sigma^2 \beta^2 \frac{n(n-1)}{4} q_0^2 - \rho^2 \sigma^2 \beta^2 \frac{n}{4} q_d^2 + \rho \mu \beta \frac{n}{2} h^2 + \frac{1}{S} \sum_i \ln Z_i \qquad (18)$$

where the partition function reads

$$Z_i = \int_{-\infty}^{+\infty} \frac{dz_i}{\sqrt{2\pi}} e^{-z_i^2/2} \int \prod_{a=1}^n dN_i^a e^{-\beta \sum_a H_{\mathrm{RS}}(N_i^a, z_i)} \; , \qquad (19)$$

with

$$-\beta H_{\mathrm{RS}}(N_i) = \frac{\beta^2 \rho^2 \sigma^2}{2} q_0 \left( \sum_a N_i^a \right)^2 + \frac{\beta^2 \rho^2 \sigma^2}{2} (q_d - q_0) \sum_a (N_i^a)^2 + \beta \sum_a (N_i^a)^2 \left( r_i + \frac{m}{K_i} \right) +$$
$$-\beta \rho \mu h \sum_a N_i^a - \beta m \sum_a N_i^a r_i - \beta \frac{r_i}{K_i} \sum_a (N_i^a)^3 - \beta(T - \lambda) \log N_i^a \; . \qquad (20)$$

At this level, the replica indices are nevertheless still coupled. To rewrite the first term in the r.h.s in a more convenient way and decouple replicas, we introduce an auxiliary Gaussian variable $z_i$ with zero mean and unit variance. Its introduction results in an additional linear contribution in $N_i$ that essentially contributes to tilting the potential. In terms of the RS Hamiltonian we eventually obtain:

$$H_{\mathrm{RS}}(N_i, z_i) = V_i(N_i) + \left[ -\frac{\beta \rho^2 \sigma^2}{2} (q_d - q_0) N_i^2 + (\rho \mu h - z_i \rho \sigma \sqrt{q_0}) N_i \right] \; . \qquad (21)$$

This result has a nice interpretation: the interaction between species leads to the emergence of an *effective* potential for a single species, and hence an effective functional response, which is species dependent. The fluctuation of the effective potential $H_{\mathrm{RS}}$ in (21) is specifically encoded in the Gaussian variable $z_i$. Since in the limit $\beta \to \infty$, $q_d \to q_0$, henceforth we shall use the notation $\Delta q = \beta \rho (q_d - q_0)$.

As we have already discussed, by tuning the parameter $m$ we can switch from a strong Allee effect, formally modeled by a double-well potential, to a weak Allee effect, which would appear more akin to a Lotka-Volterra potential, as shown in Fig. (2). What Eq. (21) shows is that due to the presence of the Gaussian variable $z_i$ the effective potential for a single given species can vary and hence change nature. For instance, $z_i$ can counterbalance the negative value of $m$ and transform a weak Allee effect into a strong one, resulting in a double-well potential rather than a single well for a finite fraction of species. Or viceversa making disappear the two stable fixed points for the species population in favor of a single one, see Fig. 3.

## 3.2 Mean-Field Replica Symmetric equations

To find the expressions for the order parameters we need to solve the integrals for the first and higher-order moments of the species abundance. In the thermodynamic limit, we can use the Laplace method and evaluate the integral by saddle-point approximation. By differentiating the action $\mathcal{A}(q_d, q_0, h)$, we end up with the following self-consistent equations for $(q_d, q_0, h)$:

$$q_d = \int \mathcal{D}z \left( \frac{\int_{N_c}^\infty dN e^{-\beta H_{\mathrm{RS}}(q_0, q_d, h, z)} N^2}{\int_{N_c}^\infty dN e^{-\beta H_{\mathrm{RS}}(q_0, q_d, h, z)}} \right) = \overline{\langle N^2 \rangle} \; , \qquad (22)$$

$$q_0 = \int \mathcal{D}z \left( \frac{\int_{N_c}^{\infty} dN e^{-\beta H_{\mathrm{RS}}(q_0, q_d, h, z)} N}{\int_{N_c}^{\infty} dN e^{-\beta H_{\mathrm{RS}}(q_0, q_d, h, z)}} \right)^2 = \overline{\langle N \rangle^2} \ , \tag{23}$$

$$h = \int \mathcal{D}z \frac{\int_{N_c}^{\infty} e^{-\beta H_{\mathrm{RS}}(q_0, q_d, h, z)} N}{\int_{N_c}^{\infty} dN e^{-\beta H_{\mathrm{RS}}(q_0, q_d, h, z)}} = \overline{\langle N \rangle} \ . \tag{24}$$

The inner average corresponds to the integration over the Boltzmann measure, while the most external one – denoted as $\mathcal{D}z \equiv \int \frac{dz}{\sqrt{2\pi}} e^{-z^2/2}$ – stands for the average over the auxiliary Gaussian variable $z$, which has specifically been introduced to decouple replicas and integrate the quenched disorder out.

As we have already discussed in Sec. (2), in order to analyze the limit of small but finite immigration rate, we introduce a cut-off in the species abundances $N_c = 10^{-2}$. This allows for an efficient numerical evaluation of the integrals. These mean-field equations are solved by the numerical procedure explained below.

---

`Algorithmic protocol`

---

- Initialize $q_d$, $q_0$, $h$ at t=0;

- Solve iteratively the equations for the three order parameters $q_d$, $q_0$, $h$ while increasing $\beta = 1/T$ with fixed cut-off $N_c = 10^{-2}$, damping $\alpha = 0.1$ and precision $\epsilon = 10^{-5}$. For instance, $h^t \leftarrow \alpha \int \mathcal{D}z \frac{\int_{N_c}^{\infty} dN e^{-\beta H_{\mathrm{RS}}(q_0^{t-1}, q_d^{t-1}, h^{t-1}, z)} N}{\int_{N_c}^{\infty} dN e^{-\beta H_{\mathrm{RS}}(q_0^{t-1}, q_d^{t-1}, h^{t-1}, z)}} + (1 - \alpha) h^{t-1}$ .

- The algorithm stops when all the parameters converge, *i.e.* when the absolute error between the $(t-1)$-value and $t$-value $\leq \epsilon$.

---

### 3.3   Generalization to the Full Replica-Symmetry Breaking (FRSB) case

The replica theory establishes that the RS scheme is correct when one single equilibrium is present. However, due to the disordered interactions the ecosystem may display several equilibria. In this case one has to use a more involved replica scheme to analyze the properties of the system [32]: the $n$ replicas are now divided in $n/m_1$ groups of $m_1$ replicas where each group of $m_1$ replicas is in turn divided in $m_1/m_2$ groups of $m_2$ replicas and so on. It is then convenient to introduce a piecewise function $q_i$, with $i = 1, 2..., k$ denoting the number of steps of broken symmetries, and to eventually take the limit $k \to \infty$. This iterative construction repeated infinitely many times gives rise to the so-called replica symmetry breaking (RSB) formalism in which $q_i$ is replaced with a continuous function

$$q(x) = q_i \qquad \text{if} \qquad m_i < x < m_{i+1} \tag{25}$$

provided the constraint

$$0 \leq m_i \leq m_{i+1} \leq 1 \ . \tag{26}$$

A one-to-one correspondence between the original piecewise function characterized by a given number of discontinuities $k$ and the parameters $q$ and $m$ can be immediately established.

Within this Ansatz, the effective Hamiltonian in Eq. (16) and the free energy become now functionals of $q(x)$, to be eventually optimized w.r.t. all possible functions $q$ having support in the unit interval, *i.e.* $\frac{\delta F[q]}{\delta q(x)} = 0$.

## 3.4 Species dependent fluctuating growth rate and differences between logistic and Allee cases

As we have explained above, the interaction with other species in the community leads to an effective potential, hence to an effective growth rate, which is species dependent:

$$H_{\mathrm{RS}}(N_i, z_i) = -\frac{\rho^2 \sigma^2}{2}\beta(q_d - q_0)N_i^2 + (\rho\mu h - z_i\rho\sigma\sqrt{q_0})N_i + V_i(N_i)\,.$$

The fluctuation due to the interactions between species is encoded in the Gaussian random variable $z_i$. Its role is to change the amplitude of the linear term in the effective potential. The minima of the effective potential correspond to the value of the abundance at which the effective per capita growth rate vanishes, *i.e.* to fixed points of population dynamics. The random contribution due to $z_i$ can then considerably affect the behavior. As we shall explain in the following, it does so in a very different way for the logistic growth and the Allee functional response. Note that even in the more complicated Full RSB case one obtains an effective potential of the same form as the one above [32]. The difference is that the random variable $z_i$ is not Gaussian. The main conclusions that we will discuss below hold also in the Full RSB case.

### 3.4.1 Lotka-Volterra logistic growth case

In the Lotka-Volterra case the potential is $V_i(N_i) = -\rho_i\left(K_i N_i - \frac{N_i^2}{2}\right)$, where $\rho_i \equiv r_i/K_i$ denotes the ratio between the single species growth rate and the carrying capacity. In this framework, the Hamiltonian in the RS approximation reads:

$$H_{\mathrm{RS}}(N_i, z_i) = \left[-\frac{\rho^2\sigma^2}{2}\beta(q_d - q_0) + \frac{\rho}{2}\right]N_i^2 + (\rho\mu h - z_i\rho\sigma\sqrt{q_0} - \rho K)N_i \qquad (27)$$

where we have assumed no species dependence on the parameters $r_i$ and $K_i$. Let us now study the minima of $H_{\mathrm{RS}}(N_i, z_i)$, which correspond to the fixed point of population dynamics in the case of zero demographic noise. Redefining $\Delta q \equiv \rho\beta(q_d - q_0)$, we eventually obtain

$$\frac{\partial H_{\mathrm{RS}}}{\partial N} = 0 \qquad \rightarrow \qquad N^*(z) = \max\left\{0, \frac{K + z\sigma\sqrt{q_0} - \mu h}{1 - \sigma^2 \Delta q}\right\}\,, \qquad (28)$$

where $z$ is, as usual, a Gaussian variable with zero mean and unitary variance. Note that the effective potential $H_{\mathrm{RS}}(N_i, z_i)$ as a function of $N_i \geq 0$ is either a monotonically increasing function from zero at $N_i = 0$ or it displays a single minimum: $z_i$ tilts the potential and leads to fluctuations between these two cases. Hence, the resulting equation for the stable fixed point is that on the r.h.s. above.

Note that if a minimum $N_2^* > 0$ exists, the fixed point $N_1^* = 0$ has always to be discarded because unstable (remember that we always work with a small but finite immigration rate). In our thermodynamic formalism this can be seen by comparing the energy of $N_1^*$ and $N_2^*$:

$$
\begin{aligned}
H_{RS}(N_1^* = 0) &= 0 \ , \\
H_{RS}(N_2^* > 0) &= -\frac{(\tilde{z} + \Delta)^2}{2(1 - \sigma^2 \Delta q)}
\end{aligned}
\tag{29}
$$

where $\Delta = \frac{K - \mu h}{\sigma \sqrt{q_0}}$. The second solution leads to a lower energy and hence has to be preferred. Therefore, in writing the resulting partition function for the Lotka-Volterra model with logistic growth, one typically enforces the solution by an Heaviside function $\theta(N^*)$.

### 3.4.2 Allee effective potential and competition between two stable fixed points

The RS effective potential in the Allee case is not just a quadratic function with a fluctuating linear term as for the logistic growth. It is instead a cubic potential with a fluctuating linear term, which gives rises to a richer phenomenology.

$$
\begin{aligned}
H_{\mathrm{RS}}(N_i, z_i) =& V_i(N_i) - \frac{\rho^2 \sigma^2}{2} \beta(q_d - q_0) N_i^2 + (\rho \mu h - z_i \rho \sigma \sqrt{q_0}) N_i = \\
=& \rho N_i^3 + N_i^2 \left[ -\frac{\rho^2 \sigma^2}{2} \beta(q_d - q_0) - r - m\rho \right] + N_i(\rho \mu h - z_i \rho \sigma \sqrt{q_0} + mr) \ ,
\end{aligned}
\tag{30}
$$

Given that the quadratic term is negative, depending on the sign of the linear term – which fluctuates because of $z_i$ – one can switch between three situations (see Fig. 3): (i) one stable fixed point at $N = 0$, (ii) two stable fixed points, one for $N_i = 0$ and one for positive abundance, (iii) only one single stable fixed point with positive abundance. The main difference with the Lotka-Volterra case is that, as shown in red in Fig. (3), for suitable values of $z_i$, the model supports three fixed points corresponding to $N^* = 0$ and

$$
N_{1,2}^* = \frac{\left( \rho \sigma^2 \Delta q / 2 + + m\rho \right) \pm \sqrt{\left( \rho \sigma^2 \Delta q / 2 + + m\rho \right)^2 - 3\rho \left( \rho \mu h + mr - z \rho \sigma \sqrt{q_0} \right)}}{3\rho}
\tag{31}
$$

If we ask the coefficient of the quadratic term to be much bigger in absolute value than the linear term, we can expand the square root and rewrite the two non-vanishing solutions as

$$
N_{1,2}^* \simeq \frac{1}{3} \left( \frac{\sigma^2 \Delta q}{2} + \frac{r}{\rho} + m \right) \pm \left( \frac{\sigma^2 \Delta q}{2} + \frac{r}{\rho} + m \right) \left[ 1 + \frac{3}{2} \frac{(z \sigma \sqrt{q_0} - \mu h - mr/\rho)}{\left( \frac{\sigma^2 \Delta q}{2} + \frac{r}{\rho} + m \right)^2} \right]
\tag{32}
$$

where we have neglected the absolute value in the second addend provided that $m$, $r$, $\rho$ are taken with positive sign. In particular, forcing the parameter $m$ to be positive we are automatically selecting a positive threshold in the population density, hence leading to a strong Allee effect.

The first solution corresponds to an unstable fixed point for the growth rate (a local maximum for the effective potential) and would read then:

$$
N_1^*(z) \simeq -\frac{1}{2} \frac{(z \sigma \sqrt{q_0} - \mu h - mr/\rho)}{\frac{\sigma^2 \Delta q}{2} + r/\rho + m}
\tag{33}
$$

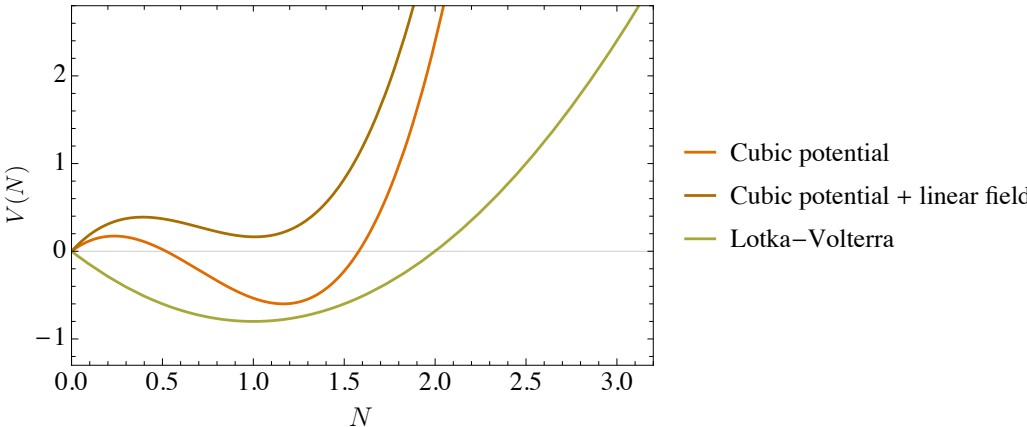

Figure 3: In the absence of interaction ($\alpha_{ij} = 0$), we compare the basic Lotka-Volterra potential with a cubic potential (in red). The effect of an external field associated to $z_i$ contribute to tilting the potential and modifying the energy barrier between minima with kind of a spinodal transition (in brown).

and, since where are interested only in positive abundances, we impose that $(z\sigma\sqrt{q_0} - \mu h - mr/\rho) < 0$. To have a well-defined solution, the external field and a threshold parameter $m$ are supposed to be sufficiently large to compensate the effect of the Gaussian variable $z$. The second non-vanishing fixed point, which corresponds to a stable fixed point for the growth rate (a local minimum for the effective potential), would instead read:

$$
\begin{aligned}
N_2^*(z) &\simeq \frac{z\sigma\sqrt{q_0}}{2\left(\frac{\sigma^2\Delta q}{2} + \frac{r}{\rho} + m\right)} - \frac{(\mu h + mr/\rho)}{2\left(\frac{\sigma^2\Delta q}{2} + \frac{r}{\rho} + m\right)} + \frac{2}{3}\left(\frac{\sigma^2\Delta q}{2} + \frac{r}{\rho} + m\right) = \\
&= \frac{\sigma\sqrt{q_0}}{2\left(\frac{\sigma^2\Delta q}{2} + \frac{r}{\rho} + m\right)}(z + \Delta) \ .
\end{aligned}
\tag{34}
$$

By substituting the solution $N_2^*(z)$ in the Hamiltonian, we would obtain

$$
\begin{aligned}
-\beta H_{RS}^{\text{Allee}} = &\left[\beta r + \beta m\rho + \frac{\beta\rho\sigma^2}{2}(\rho\beta(q_d - q_0))\right]\left[\frac{\sigma\sqrt{q_0}}{2\left(\frac{\sigma^2\Delta q}{2} + r/\rho + m\right)}(z + \Delta)\right]^2 + \\
&- \beta\rho\left[\frac{\sigma\sqrt{q_0}}{2\left(\frac{\sigma^2\Delta q}{2} + r/\rho + m\right)}(z + \Delta)\right]^3 - \beta\rho\left(\mu h - z\sigma\sqrt{q_0} + \frac{rm}{\rho}\right)
\end{aligned}
\tag{35}
$$

The major difference with the logistic growth case is that, when the solutions $N_2^*(z)$ and $N^* = 0$ exist, one has to compare their energies to select the one that dominates for vanishing demographic noise. The latter has zero energy. As a consequence, to favour the second solution one should have the former with a lower energy, which leads to a second-order inequality in

$z + \Delta$ :

$$\left(\frac{r}{\rho} + m + \frac{\sigma^2 \Delta q}{2}\right) \frac{\sigma \sqrt{q_0}(z+\Delta)}{2\left(\frac{\sigma^2 \Delta q}{2} + \frac{r}{\rho} + m\right)} - \frac{\left[\sigma \sqrt{q_0}(z+\Delta)\right]^2}{4\left(\frac{\sigma^2 \Delta q}{2} + \frac{r}{\rho} + m\right)^2} - \left(\mu h - z\sigma \sqrt{q_0} + \frac{rm}{\rho}\right) \geq 0 \tag{36}$$

At variance with the simple Lotka-Volterra case with logistic growth, the introduction of a cubic potential makes the competition between the two fixed points not trivial. Note that if the coefficient of the linear term, function of the Gaussian variable $z$ and the generalized external field, is sufficiently large, the second finite solution appears to be sub-leading compared to the one corresponding to extinction.

## 4   Phase diagrams with weak and strong Allee effect

In the following, we obtain the phase diagrams of the model with a cubic potential as in Eq. (2), random symmetric interactions between species and small but finite demographic noise. In Fig. (4) we specifically highlight two distinct phases in the presence of demographic noise

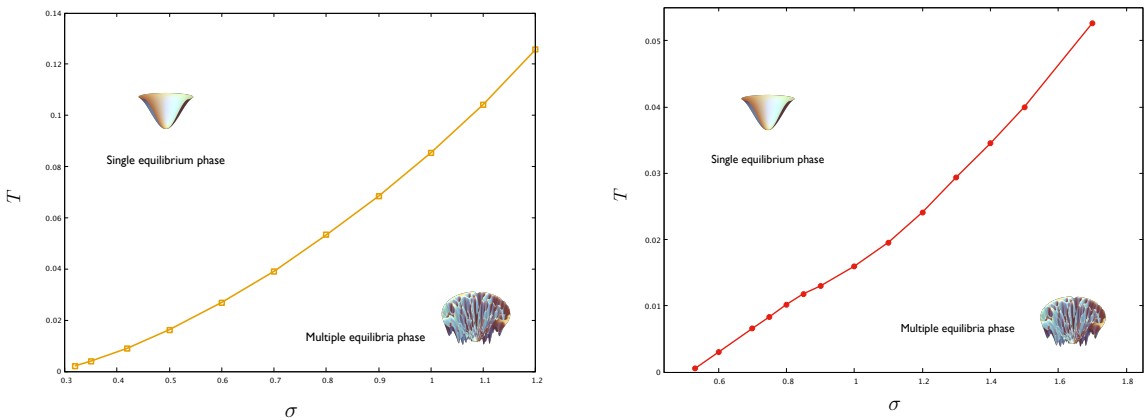

Figure 4: Phase diagram showing the amplitude of the demographic noise $T$ as a function of the heterogeneity parameter $\sigma$ for the strong Allee ($\mu = 5$ and $m = 1$) **on the left** and for the weak Allee effect ($\mu = 5$ and $m = -0.5$) **on the right**.

and small but non-zero immigration. The two phase diagrams, respectively for the strong and the weak Allee effect, have been obtained at fixed average interaction $\mu$ since we observed no sensitive dependence on it[4]. For sufficiently high demographic noise (corresponding to the high-temperature phase) only a single equilibrium phase is detectable: the noise is so strong that interactions among species do not play any significant role. Conversely, at low demographic noise or highly heterogeneous interactions, such single equilibrium phase becomes unstable leading to a multiple equilibria regime[5]. In the following we will show how this second phase can be mathematically described within a Full RSB Ansatz in the replica formalism.

---

[4]A similar outcome was pointed out also for the Lotka-Volterra model with finite demographic noise [11].

[5]The orange and red lines in the two diagrams correspond to a vanishing replicon mode of the Hessian matrix of the free energy, as better explained in the next Section.

This aspect has important consequences on the organization of equilibria, which have the same organization as stable thermodynamic states in mean-field spin glasses [32].

In the following we will first analyze the transition from the one equilbrium phase to the multiple equilibria phase. Once the phase diagram is well established down to vanishing demographic noise, we will then proceed with the analysis of the nature of low-demographic noise (low-temperature) phase. We will come back to this point later, precisely in Sec. (4.2.1).

## 4.1 Transition from the one equilibrium phase to the multiple equilibria phase: massless modes and diverging susceptibilities

The emergent multiple equilibria phase in the low-demographic noise regime can be understood using concepts rooted in statistical physics of disordered systems. As we shall show, the multiple equilibria phase in Fig.(4) is critical and hence associated with strong correlations among the degrees of freedom, hence leading to diverging response functions. Our first aim is to obtain the condition for the transition between the two phases. We will also obtain a general condition for stability. This will allow us to show later that the multiple equilibria phase is marginally stable and hence critical.

To analyze correlations and fluctuations, let us consider the three following correlation functions, which will be useful also in the following:

$$
\begin{aligned}
\mathcal{C}_1 &= \overline{\langle N_i N_j \rangle^2} - \overline{\langle N_i \rangle^2} \cdot \overline{\langle N_j \rangle^2}, \\
\mathcal{C}_2 &= \overline{\langle N_i N_j \rangle \langle N_i \rangle \langle N_j \rangle} - \overline{\langle N_i \rangle^2} \cdot \overline{\langle N_j \rangle^2}, \\
\mathcal{C}_3 &= \overline{\langle N_i \rangle^2 \langle N_j \rangle^2} - \overline{\langle N_i \rangle^2} \cdot \overline{\langle N_j \rangle^2}
\end{aligned}
\tag{37}
$$

from which we have subtracted the product of the single averages to focus only on the connected part of the correlation function. From these definitions one can introduce two special combinations of such correlations, the *spin-glass susceptibility* and the *non-linear susceptibility*, *i.e.* $\chi^{\mathrm{SG}} = \mathcal{C}_1 - 2\mathcal{C}_2 + \mathcal{C}_3$ and $\chi^{\mathrm{NL}} = \mathcal{C}_1 - 4\mathcal{C}_2 + 3\mathcal{C}_3$. However, for our purposes, we will focus only on the first instance of susceptibility.

In the same spirit, one can also generalize the definition to embed the time-dependence in the correlator, $C(t, t')$, and look at fourth-order correlation function corresponding in jargon to $\chi_4(t, t')$. It can be shown that in the single-equilibrium phase, where the dynamics tends to be stationary after a short transient, this quantity is nothing but the the spin-glass susceptibility in the long-time limit:

$$
\chi_{\mathrm{SG}} = \frac{1}{S \overline{\langle N_i^2 \rangle_c}^2} \sum_{ij} \left( \langle N_i N_j \rangle - \langle N_i \rangle \langle N_j \rangle \right)^2 ~,
\tag{38}
$$

which is appropriately normalized to the connected part of the self-correlation and the number of species $S$ [7]. Recalling the definition of the standard susceptibility $\chi_{ij} = \langle N_i N_j \rangle - \langle N_i \rangle \langle N_j \rangle$, one can immediately figure out the connection in between, *i.e.* $\chi_{\mathrm{SG}} = \overline{\chi_{ij}^2}$. Remarkably, such a definition of susceptibility is strictly linked to the correlations of the overlap matrix as it can be immediately proven by introducing a replicated effective field theory. The fluctuations contributing to the spin-glass correlation functions are due to a special mode of the stability matrix, the so-called *replicon mode*, whereas those contributing to the nonlinear correlation function are related to the so-called *longitudinal-anomalous* modes. To investigate stability properties of the different phases, we thus introduce the Hessian matrix of the free energy,

which allows us to analyze the harmonic fluctuations in terms of $\delta Q_{ab}$. Thanks to symmetry group properties in the replica space, the diagonalization of the stability matrix can be expressed only in terms of three different sectors. Following [34], we define three eigenvalues: the *longitudinal*, $\lambda_{\mathrm{L}}$, the *anomalous*, $\lambda_{\mathrm{A}}$, and the *replicon*, $\lambda_{\mathrm{R}}$. Since we are specifically interested in possible Replica Symmetry Breaking (RSB) effects, we focus on the replicon mode. By contrast, a zero longitudinal mode can give information in terms of spinodal points describing how a state opens up along an unstable direction and originates then a saddle. The detection of a vanishing replicon mode from the sufficiently high demographic noise (single equilibrium) phase is related to the appearance of *marginal states*, which turn out to be extremely relevant also because of their intimate connection with out-of-equilibrium aging dynamics [35–37].

We consider then the variation of the RS action with respect to the overlap matrix

$$\mathcal{A}(Q_{ab}, Q_{aa}, H_a) = -\rho^2\sigma^2\beta^2 \sum_{a<b} \frac{Q_{ab}^2}{2} - \rho^2\sigma^2\beta^2 \sum_a \frac{Q_{aa}^2}{4} + \rho\mu\beta \sum_a \frac{H_a^2}{2} + \frac{1}{S} \sum_i \ln Z_i \ , \quad (39)$$

for which we recall also the expression of the partition function:

$$Z_i = \int \prod_a dN_i^a e^{\left[ \frac{\beta^2\rho^2\sigma^2}{2} \sum_{a<b} Q_{ab}N_i^a N_i^b + \beta^2\rho^2\sigma^2 \sum_a (N_i^a)^2 \frac{Q_{aa}}{2} - \rho\beta\mu \sum_a N_i^a H^a - \beta V_i(N_i^a) - \ln N_i^a \right]} . \quad (40)$$

By differentiating the action with respect to the overlap matrix, we obtain to the first order:

$$-\frac{\partial A}{\partial Q_{ab}} = \beta^2\rho^2\sigma^2 Q_{ab} - \beta^2\rho^2\sigma^2 \overline{\langle N^a N^b \rangle} \ , \quad (41)$$

and to the second order

$$\mathcal{M}_{abcd} \equiv -\frac{\partial^2 \mathcal{A}}{\partial Q_{ab} \partial Q_{cd}} = \beta^2\rho^2\sigma^2 \left[ \delta_{(ab),(cd)} - (\beta^2\rho^2\sigma^2) \overline{\langle N^a N^b, N^c N^d \rangle_c} \right] \ , \quad (42)$$

where the subscript $\langle \cdot \rangle_c$ always stands for the connected part of the correlator. We can now take advantage of the underlying symmetry in the replica space and decompose the stability matrix (42) as follows

$$\mathcal{M}_{abcd} = M_{ab,ab} \left( \frac{\delta_{ac}\delta_{bc} + \delta_{ad}\delta_{bc}}{2} \right) + M_{ab,ac} \left( \frac{\delta_{ac} + \delta_{bd} + \delta_{ad} + \delta_{bc}}{4} \right) + M_{ab,cd} \quad (43)$$

whose projection into the replicon subspace yields

$$\lambda_{\mathrm{R}} = (\beta\rho\sigma)^2 \left[ 1 - (\beta\rho\sigma)^2 \overline{(M_{ab,ab} - 2M_{ab,ac} + M_{ab,cd})} \right] . \quad (44)$$

Finally, the three contributions appearing in the second term of Eq. (44) can be re-expressed as

$$M_{ab,ab} - 2M_{ab,ac} + M_{ab,cd} = \left[ \langle (N^a)^2 (N^b)^2 \rangle - 2\langle (N^a)^2 N^b N^c \rangle + \langle N^a N^b N^c N^d \rangle \right] . \quad (45)$$

In the simplest scenario corresponding to the presence only of a single equilibrium, the replicon mode can be eventually written as

$$\lambda_{\mathrm{R}} = (\beta\rho\sigma)^2 \left[ 1 - (\beta\rho\sigma)^2 \overline{(\langle N^2 \rangle - \langle N \rangle^2)^2} \right] \ , \quad (46)$$

where the averaged difference describes the fluctuations between the first and second moments of the species abundances within one state, namely between the diagonal value $q_d$ and the off-diagonal contribution $q_0$ of the overlap matrix. By computing $\lambda_R$ for the one equilibrium phase and by detecting the points in which it vanishes we have obtained the transition lines shown in Fig.(4). In the following sections we shall investigate the nature of the multiple equilibria phase.

## 4.2 FRSB nature of the multiple equilibria phase.

In the following, we will first establish that the transition to the multiple equilibria phase is continuous and then prove that the transition is towards a FRSB phase.

### 4.2.1 Continuous transition and computation of the breaking point

We will analyze the replica symmetry breaking solution corresponding to the multiple equilibria phase. We will present a method to derive the equation for the breaking point $x^*$, *i.e.* the point where $q(x)$ deviates from the constant profile with $\dot{q}(x^*) \neq 0$. This can be done by performing a perturbative expansion close to the critical line (in orange and red respectively in Fig. 4), a.k.a the de Almeida Thouless (dAT) line in spin glass literature, which allows us to determine the nature of the emerging transition depending on whether $x^*$ satisfies a specific criterion or not. Precisely, if $x^* \in [0, 1]$, the instability of the RS solution gives rise to a continuous transition towards a RSB phase, which can either be FRSB or $k$-RSB with finite $k$. Conversely, if $x^* \notin [0, 1]$, the expansion around the dAT line turns out to be unphysical: this means that the transition must be anticipated by a discontinuous RSB transition. Finding out which kind of instability takes place at the transition is very important to determine the nature of the multiple equilibria phase.

To determine if $x^* \in [0, 1]$ one typically consider the free energy functional $f[q(x); q_d]$ where $q_d$ is the diagonal value and $q(x)$ is function of the continuous variable $x \in [0, 1]$. By performing a perturbative expansion of the free energy, one can then compute the breaking point $x^*$ in whose vicinity $q(x)$ becomes nontrivial. Therefore, using only information from the RS calculation, one can claim on the universality class of the emerging transition. The interested reader will find more details in Appendix A and an extensive proof in [38] (see also references within).

The standard strategy is to consider the functional derivative of the free energy and use the following relations:

$$\frac{d}{dx}\frac{\delta f[q(x); q_d]}{\delta q(x)} = f^{(2)}[q(x); q_d]\dot{q}(x) \tag{47}$$

$$\frac{d}{dx}f^{(n)}[q(x); q_d] = f^{(n+1)}[q(x); q_d]\dot{q}(x) \qquad \text{for} \quad n \geq 2 \ . \tag{48}$$

Accordingly, the second derivative of the free energy can be written as

$$f^{(2)}[q(x); q_d]\dot{q}(x) = \frac{1}{2}\frac{d}{dx}\int dh \, P(h, x)f'(x, h)^2 = \frac{\dot{q}(x)}{2}\int dh P(h, x)f''(x, h)^2 \ . \tag{49}$$

$P(h, x)$, formally introduced as a Lagrange multiplier, has a deeper physical meaning in the spin-glass literature encoding the distribution of local magnetic fields within metabasins at level $x$ according to a hierarchical organization of states. Then, one should consider the variations of the above expressions in the space of all independent functions.

According to Eq. (49), if there exists an interval $\in [0,1]$ above which $\dot{q}(x) \neq 0$, one can further simplify the expression

$$f^{(2)}[q(x); q_d] = \frac{1}{2} \int dh P(h,x) f''(x,h)^2 \ , \tag{50}$$

which is equivalent to determining the vanishing value of the replicon mode of the stability matrix, *i.e.* the marginal stability condition for the RS solution. Hence, the breaking point can be obtained in a straightforward way by the differentiation of the above equation with respect to $x$ (see again [38]):

$$\frac{d}{dx} \int dh P(h,x) f''(x,h)^2 = \dot{q}(x) \left[ dh P(h,x) f'''(x,h)^2 - 2x \int dh P(h,x) f''(x,h)^3 \right] \ . \tag{51}$$

Taking advantage of the variational equations (47)-(48), and imposing $f^{(3)}[q(x); q_d] = 0$, one obtains:

$$\int dh P(h,x) f'''(x,h)^2 - 2x \int dh P(h,x) (f''(x,h))^3 = 0 \ , \tag{52}$$

which in terms of the breaking point $x^*$ yields

$$x^* = \frac{dh P(h,x) f'''(x,h)^2}{2 \int dh P(,hx) f''(x,h)^3} \ . \tag{53}$$

According to the obtained value of $x^*$ along the critical line, two options are possible:

- if $x^* \in [0,1]$, a continuous transition to a RSB phase takes place. This can occur either towards a full RSB or a $k$-RSB transition with finite $k$;

- if $x^* \notin [0,1]$, the perturbative expansion is associated with an unphysical solution. The only reasonable scenario corresponds to the fact that the transition must be anticipated by another kind of instability, typically discontinuous in the order parameter.

Starting from Eq. (53) and rephrasing the free energy in terms of the $N$s variables for the species abundances, the final condition for the breaking point corresponds to computing the ratio between the third and the second cumulants:

$$x^* = \frac{\left( \overline{\langle N^3 \rangle} - 3\overline{\langle N^2 \rangle \langle N \rangle} + 2\overline{\langle N \rangle^3} \right)^2}{2 \left( \overline{\langle N^2 \rangle} - \overline{\langle N \rangle^2} \right)^3} \tag{54}$$

where, as before, the brackets $\langle \cdot \rangle$ stand for the average over the species abundance distribution while the upper bar $\bar{\cdot}$ represents the average over the quenched disorder. In Fig. (5) we reproduce its trend as a function of noise amplitude for the (weak) Allee effect. Similar results are obtained for the strong Allee effect. The conclusion of our analysis is that the multiple equilibria phase is characterized by the first option above, *i.e.* $x^* \in [0,1]$ and hence a continuous transition to a RSB phase. We will show below that it is a FRSB phase.

In the Appendix we also present an alternative way to derive the breaking point, in line with the computation performed in [39].

### 4.2.2   Full RSB phase

To exclude one of the two scenarios that are linked to the case $x^* \in [0,1]$, we need to go more in detail and obtain the resulting expression also for the slope. We assume that $q(x)$ is analytical in the vicinity of the phase transition and consider the following equation

$$\frac{d}{dx}\int dh P(h,x)f''(x,h)^2 = \dot{q}(x)\left[dhP(h,x)f'''(x,h)^2 - 2xdhP(h,x)f''(x,h)^3\right] \tag{55}$$

We differentiate this Eq. with respect to $x$ to obtain:

$$\frac{d}{dx}\left[\int dhP(h,x)f'''(x,h)^2 - 2x\int dhP(h,x)f''(x,h)^3\right] = \\ \dot{q}(x)\int dh\, P(h,x)Q_4(x,h) - 2\int dh\, P(h,x)f''(x,h)^3 \tag{56}$$

where $Q_4(x,h)$ denotes

$$Q_4(x,h) = f''''(x,h)^2 - 12x^*f''(x,h)f'''(x,h)^2 + 6x^{*2}f''(x,h)^4 \,. \tag{57}$$

From Eq. (56), we eventually derive:

$$\dot{q}(x) = \frac{2\int dhP(h,x)f''(x,h)^3}{\int dhP(h,x)Q_4(x,h)} \tag{58}$$

which, once re-expressed in terms of the relative species abundances, coincides with the ratio between the second order moment and the quartic cumulant, $i.e.$:

$$\dot{q} = \frac{2\left(\overline{\langle N^2\rangle} - \overline{\langle N\rangle^2}\right)^3}{2\left[\overline{\langle N^4\rangle}_{\text{cum}}^2 - 12x^*\left(\overline{\langle N^2\rangle} - \overline{\langle N\rangle^2}\right)\left(\overline{\langle N^3\rangle} - 3\overline{\langle N^2\rangle\langle N\rangle} + 2\overline{\langle N\rangle^3}\right)^2 + 6(x^*)^2\left(\overline{\langle N^2\rangle} - \overline{\langle N\rangle^2}\right)^4\right]} \tag{59}$$

where we have denoted as $\langle N^4\rangle_{\text{cum}}$ the expression for the quartic cumulant. If the value of the slope at the breaking point $x^*$ is positive, we can state that the resulting transition is continuous towards a FRSB phase. In the opposite case, it would be described by a non-marginal 1RSB solution. We have verified numerically that it is the former case that is realized both for weak and strong Allee effect. Thus, the multiple equilibria phase that emerges at low demographic noise is a FRSB one.

In conclusion, contrary to the Lotka-Volterra case with logistic growth, for which we demonstrated the existence of a 1RSB transition to be eventually replaced by an amorphous Gardner phase in the zero-demographic noise limit [11], here we find no evidence of such a phase. We do not observe any jump in $x^*$ and the corresponding slope at the breaking point keeps a positive-definite sign as a signature of the fact that the first non-trivial solution implies an infinite number of symmetry breaking. As the demographic noise is sufficiently low, the RS solution is automatically broken into a FRSB marginal phase. This shows a major difference between the logistic growth and the Allee cases, which therefore display very different multiple equilibria phases. Let us discuss below what are the main features that distinguish them.

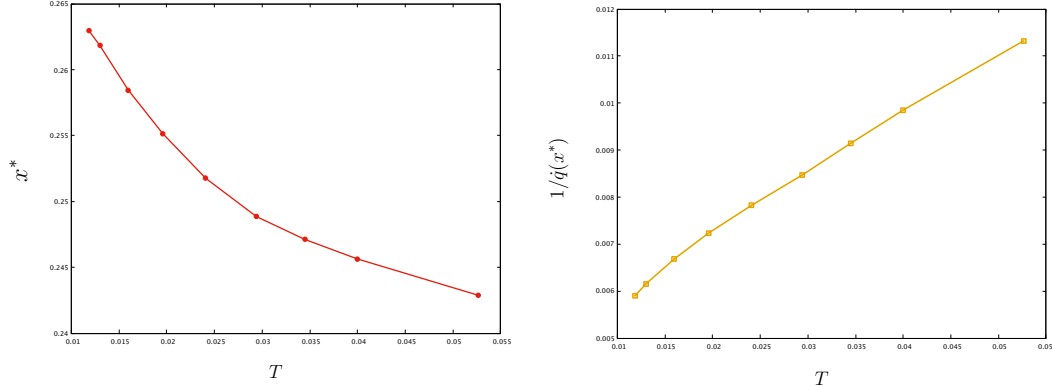

Figure 5: Breaking point $x^*$ (left) and inverse slope $1/\dot{q}(x^*)$ (right) versus temperature $T$ represented for the *weak Allee effect*, in red in Fig. 4. The plots are obtained at $\mu = 5$ and $m = -0.5$. The breaking point is definitely smaller than one, and the slope evaluated at $x^*$ remains positive as a signature of a continuous transition to a FullRSB phase.

## 4.3 Properties of the multiple equilibria phase

In the case of mean-field glassy systems one can give a precise meaning to the free-energy landscape. The number of minima, and more generally of critical points of given index, has been computed and analyzed thoroughly [40–42], and even more rigorously in the last years [43]. Such works established the existence of two main universality classes, which are associated with different thermodynamic and dynamical properties:

- **Spin-glass models** characterized by sub-exponential number of free-energy minima in the system size. The free-energy barriers are expected to be sub-extensive and the minima organized in a hierarchical fractal structure [32].

- **Structural glass models** wherein the number of free-energy minima is exponential in the system size. At variance with the previous universality class, free-energy barriers are extensive and typically a low-noise dynamics starting from random initial conditions remains stuck in high free-energy minima (called *threshold states*).

In [11] we have proven that it is precisely the second class that characterizes the multiple equilibria phase of the Lotka-Volterra model with logistic growth. Here, we show that for functional responses displaying an Allee effect the ecosystem belong to the *first class* of models. In this case, the resulting picture is thus reminiscent of mean-field spin glasses for which, according to the Parisi solution [44,45], an infinite number of pure ultrametric states emerges.

## 5 Marginal stability and pseudo-gap distribution of the local curvatures

In the following we wish to link the marginal stability of the FRSB phase (a general result in the context of replica theory [32]) to a pseudo-gap distribution of the local curvatures of the effective potentials for the species. This is the generalization to our case of the pseudo-gap for instantaneous local field distribution of mean-field spin glasses [32].

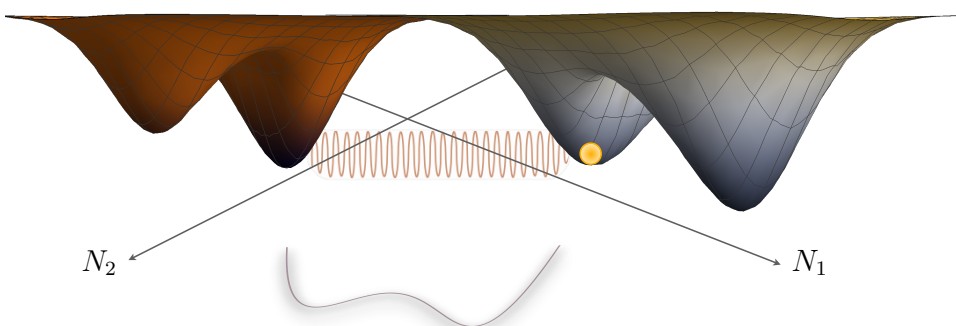

Figure 6: Pictorial representation of two interacting double-well potentials joined by a fictitious spring. Because of the interaction, the minima of the resulting 2-dimensional potential, as projected on the plane $(N_1, N_2)$, are located in completely different positions than the 3-dimensional starting potentials.

## 5.1 Marginal stability at vanishing demographic noise: the replicon condition

As we have already stressed, it is known from the replica theory that the FRSB phase is marginal, which means that the Hessian of the replicated free-energy displays a zero eigenvalue (the replicon). As shown in [7], one finds that[6]:

$$\lambda_{\text{repl}} \propto 1 - \overline{\frac{\rho^2 \sigma^2}{\left(-\Delta q \rho^2 \sigma^2 + V_i''(N_i^*)\right)^2}} \,, \tag{60}$$

where $V_i''(N_i^*)$ is the curvature of the potential associated with the functional response of species $i$ evaluated at the abundance of that species, which is associated to the global minimum of the effective potential for that species. The overline denotes the average over the species. This relation is valid for any equilibrium both in the single and multiple equilibria phase.

By introducing the notation $V_{\text{eff}}(N_i)$ for the effective potential for species $i$, we find that the curvature of the effective potential evaluated at the global minimum is precisely $-\Delta q \rho^2 \sigma^2 + V_i''(N_i^*)$. Therefore, the stability condition for a given phase (*i.e.* a positive replicon eigenvalue) reads:

$$1 - (\rho\sigma)^2 \int \frac{dV_{\text{eff}}'' \, P(V_{\text{eff}}''(N^*))}{\left(V_{\text{eff}}''(N^*)\right)^2} \geq 0 \tag{61}$$

where we have introduced the distribution of the $V_{\text{eff}}''(N^*)$ in a given equilibrium. This inequality implies that a necessary condition for stability is to have a pseudo-gap in $P(V_{\text{eff}}''(N^*))$ otherwise the integral would diverge and the condition would not be satisfied. More specifically, by assuming that the probability behaves as $P(V_{\text{eff}}'') \sim |V_{\text{eff}}''|^\alpha$ at small argument, stability implies that $\alpha > 1$. Marginally stable behavior corresponds to the choice $\alpha = 1$ with logarithmic corrections to the power law, which makes the integral convergent. The value $\alpha = 1$

---

[6] A more detailed analysis in the limit of zero demographic noise can also be found in Appendix C.

implies that the smallest attainable curvature $V''_{\min}$ is such that

$$N \int_0^{V''_{\min}} P(V''_{\min}) \, dV''_{\min} \sim O(1) \rightarrow V''_{\min} \sim \frac{1}{\sqrt{N}} \ . \tag{62}$$

This derivation of the pseudo-gap relies on the expression of the replicon in terms of the distribution of local curvatures. As done for spin-glasses, one can establish the same result using general arguments without referring to the replica method. We will now show how this can be done in two different complementary ways. In Ising models one is typically interested in identifying the probability distribution of the effective fields in the zero-temperature phase. The seminal work by Anderson, Palmer and coworkers [46, 47] is indeed based on energy considerations after a spin flip on all the $n$ sites that are subject to the lowest fields and allows one to claim that $P(h) \sim h^a$ as $h \rightarrow 0$, with exponent $a \geq 1$. This argument has been supported by numerical simulations corroborating the hypothesis of a linear dependence in $h$ at low fields. In the following, we will proceed in a similar way studying local excitations.

## 5.2 First argument: dynamical cavity

There are different possible ways to determine the exponent $\alpha$. In the following we shall propose two possible directions: the former is inspired by a dynamical cavity approach, while the second one is based on a Gaussian approximation for dealing with coupled double-well potentials. Let us start with the equation for the time evolution of the species abundances:

$$\dot{N}_i(t) = N_i \left[ -V'(N_i) - \sum_{j \neq i} \alpha_{ij} N_j + \xi_i \right] \ . \tag{63}$$

By plugging a new species $k$ in the ecosystem, this equation becomes:

$$N_i = N_{i \setminus k} - \frac{\partial N_i}{\partial \xi_i} \alpha_{ik} N_k \tag{64}$$

where the second contribution represents the product between the diagonal susceptibility $\chi_{ii}$ and the so-called *cavity field*. There are then two possibilities: i) either $N_k = 0$; ii) or $N_k \neq 0$. In this second case:

$$\begin{aligned}
&- V'(N_k) - \sum_{i \geq 1} \alpha_{ki} N_i + \xi_k = \\
&- V'(N_k) - \sum_i \alpha_{ki} N_{i \setminus k} + N_k \sum_i \alpha_{ki} \alpha_{ik} \frac{\partial N_i}{\partial \xi_i} + \xi_k \ ,
\end{aligned} \tag{65}$$

which, once differentiated w.r.t $\xi_k$, leads to:

$$-V''(N_k) \frac{\partial N_k}{\partial \xi_k} + \frac{\partial N_k}{\partial \xi_k} \sum_i \alpha_{ki} \alpha_{ik} \chi_{ii} \ . \tag{66}$$

If we consider its average contribution, we also get:

$$\left\langle \frac{\partial N_k}{\partial \xi_k} \right\rangle = \left\langle -\frac{1}{V''(N_k) - \sum_i \alpha_{ki} \alpha_{ik} \chi_{ii}} \right\rangle \tag{67}$$

basically meaning that the introduction of a new variable $N_k$ corresponds to a perturbation on species $N_i$ at a given time $t$. We nevertheless ask $N_i(t)$ to remain of $\mathcal{O}(1)$. Since the quenched variables fluctuate keeping their variance $\alpha_{ik}^2 \sim \frac{1}{N}$ and the variable $N_k$ must be always $\mathcal{O}(1)$, the cavity field must satisfy the following condition: $\alpha_{ik} N_k \sim \frac{1}{\sqrt{N}}$. By means of a scaling argument we can conclude that

$$\frac{1}{V''_{\text{eff}}(N^*)} \frac{1}{\sqrt{N}} \sim \mathcal{O}(1) \ , \tag{68}$$

which implies a power-law behavior for the effective potential with

$$V''_{\text{eff}}(N^*) \sim \frac{1}{\sqrt{N}} \ . \tag{69}$$

As a consequence, the relationship in (62) is satisfied for $\alpha \geq 1$.

## 5.3  Second argument: Gaussian model of coupled potentials

We now follow closely an argument that has been proposed for spin glasses [32]. We assume that all single-species excitations are stable – which means that all local curvatures are strictly positive – and ask what is the lowest typical values of the local curvatures that the ecosystem can display in order to maintain stability when multi-species excitations are taken into account.

We focus on two coupled potentials and study the resulting stability condition given by their interaction, as shown in Fig. (6). We assume that since the interaction is very small the change in the values of the equilibrium abundances $N_i^*$ and $N_j^*$ can be neglected. Since we want to study the stability, we consider the energy governing the two interacting species valid for small fluctuations:

$$\mathcal{E}_{ij} = \frac{1}{2} N_i^2 V''_{\text{eff}}(N_i^*) + \frac{1}{2} N_j^{*2} V''_{\text{eff}}(N_j) + \alpha_{ij} N_i N_j \tag{70}$$

which leads to a Hessian

$$\mathbf{A} = \begin{pmatrix} V''_{\text{eff}}(N_i^*) & \alpha_{ij} \\ \alpha_{ij} & V''_{\text{eff}}(N_j^*) \end{pmatrix} \tag{71}$$

In the same spirit of magnetic systems, if $\alpha_{ij} < 0$ energy tends to decrease only if the two variables are aligned with $N_i \sim N_j$. Conversely, if $\alpha_{ij} > 0$ energy decreases with $N_i \sim -N_j$ exactly as it happens for antiferromagnetic systems. Stability requires positive eigenvalues and hence

$$V''_{\text{eff}}(N_i^*) V''_{\text{eff}}(N_j^*) - \alpha_{ij}^2 > 0 \ . \tag{72}$$

A distribution $P(V''_{\text{eff}})$ with a pseudogap with exponent $\alpha$ implies that the minimal $V''_{\text{eff}}$ that one can find in the system are of the order $N^{-\frac{1}{1+\alpha}}$. Given that $\alpha_{ij} \sim 1/\sqrt{N}$, for $\alpha < 1$ the couple $i, j$ associated with the two smallest values of $V''_{\text{eff}}$ would become necessarily unstable due to interactions, hence implying that the ecosystem as a whole cannot be stable if $\alpha < 1$. Therefore, stability requires $\alpha \geq 1$, with the value $\alpha = 1$ corresponding to marginal stability. As for spin-glasses, one finds a similar result also considering excitations of more than two particles.

Note that for the argument above we used the assumption that the changes in the value of the equilibrium abundances $N_i^*$ and $N_j^*$ due to the interaction can be neglected. This is indeed correct as far as $\alpha > 1$, as it can easily checked. A more detailed argument can be found in Appendix D.

# 6 Summary of the results: Allee versus Logistic Growth case

In the following, we will briefly recap the main differences we have found in this work between the behavior of the Lotka-Volterra model with logistic growth and with the Allee effect.

- **Different Phase Diagram** In terms of the emerging phase diagram in the two-dimensional plane $(T, \sigma)$, the Lotka-Volterra model turns out to be characterized by three different phases: i) a single equilibrium phase (formally described by a RS solution); ii) a multiple equilibrium regime (well-defined within a 1RSB Ansatz) with an exponential number of equilibria in the system size; iii) a marginally stable Gardner phase. All these regimes have been studied in detail in [11]. The resulting scenario is completely different in the Allee model, as shown in Fig. (4). In this case, only two phases take place both for the weak and the strong Allee effect: i) a single equilibrium phase; ii) a multiple equilibria regime, which is well-defined within a Full-RSB (FRSB) Ansatz, with no signature of a Gardner phase. Furthermore the transition between the single and multiple equibria phase is *continuous.*

- **Different Nature of the Multiple Equilibria Phase** In the Allee case, the multiple stable equilibria are the ones characteristic of the FRSB phase: they are not exponential in number, they are separated by sub-extensive barriers and organized in a hierarchical way, as equilibrium states of mean-field spin glasses [32]. This is very different from the logistic growth case in which the number of equilibria is exponential and the barriers are extensive [11].

- **Pseudogap distribution of the local curvatures** Another extremely relevant feature relies on the marginal stability condition as shown in Eq. (60). The Lotka-Volterra model with logistic growth is rather unique in the sense that its response function does not depend on $N^*$, hence the local curvatures do not fluctuate. As soon as one consider a functional response which is not a linear function, local curvatures fluctuate. In the multiple equilibria phase their distribution is characterized by a pseudo-gap which is a smoking-gun signature of marginal phases.

# 7 Conclusion

In this work, we have studied a disordered mean-field model that specifically allows us to reproduce the main properties of the so-called Allee effect in theoretical ecology and to unveil the complex structure of the equilibria in the energy landscape. The Allee effect, which results in a positive correlation between population density and mean individual fitness, can occur essentially in two different ways – hence defined as *strong Allee effect* and *weak Allee effect* – depending on whether there exists an initial population threshold below which the population goes extinct. In its strong variant it turns out to be characterized by the appearance of a new stable fixed point at positive abundance competing with the one in $N^* = 0$ corresponding to extinction in absence of immigration and colonization. This underlying bistability has interesting connections with bistable networks and hysteresis effects in disordered systems. Our outcomes can therefore lead to wide-ranging applications in population dynamics, genetics, epidemiology up to cancer evolution thanks also to recent advances in modern engineering and synthetic biology.

Formally, our work relies on introduction of a cubic potential in the species abundances, which turns out to be particularly convenient not only to extend well-known predictions based on the Malthusian principle and logistic function for the mean population growth, but also to go beyond very recent results obtained for the Lotka-Volterra logistic growth model in the well-mixed assumption. Notably, we have established the full phase diagram in the presence of random symmetric interactions between species and finite demographic noise. By varying the amplitude of the demographic noise together with the heterogeneity parameter of the interaction matrix, we have highlighted two phases with a different underlying symmetry. Then, by taking advantage of a perturbative calculation, we have investigated and distinguished a single equilibrium phase, where the phase space is purely ergodic, from a multiple equilibria regime, where the symmetry is broken infinitely many times, thus associated with a rough landscape structure. A main result is that the phase diagram as well as the nature of the multiple equilibria phase differ substantially from the ones obtained in the case of logistic growth, thus unveiling the relevance of the species functional response in determining the aggregate properties of the entire ecosystem. Then, one can wonder what would be the effect of adding a small asymmetry to the interactions. Theoretical predictions have been discussed in a close-related framework for a purely competitive environment with logistic growth [11]. In the case of the Allee effect, we expect that the multiple equilibria FRSB phase is replaced by a chaotic dynamical phase that corresponds to a slow *surfing* on top of the marginally stable states [11, 48].

Finally, we have presented a novel prediction in terms of fluctuations of the effective potential (or effective growth rate) for single species in a large interacting ecosystem. Marginality of the multiple equilibria phase leads to a pseudo-gap in the distribution $P(V_{\text{eff}}''(N^*))$ of the local curvatures, a phenomenon that have attracted a lot of attention in the physics of marginal states of matter [27].

**Acknowledgments**   We acknowledge discussions with G. Bunin, S. De Monte and P. Urbani.

**Funding information**   This work was supported by the Simons Foundation Grant on Cracking the Glass Problem (#454935 Giulio Biroli). A.A. acknowledges support by the L'Oréal UNESCO Young Talents France Fellowship.

## A   Derivation of the breaking point

In Sec. 4.2.1 we showed the final expression for the breaking point allowing us, together with the slope $\dot{q}(x)$, to distinguish between a discontinuous and a continuous phase transition in the order parameter. This provide a crucial information to better investigate the properties of the equilibria in the free-energy landscape and argue about the resulting transitions. The computation of these two quantities allows us to avoid a too convoluted formalism, which would in principle require the stability computation of multiple steps of replica symmetry breaking. In the following, we will then discuss the main passages to obtain the expressions for the breaking point and the slope in a very general setting, as shown in [38].

We start from the expression of the functional derivative of the free energy

$$\frac{\delta f[q(x); q_d]}{\delta q(x)} = \frac{1}{2} \int dh \, P(h, x)(f'(h, x))^2 \ . \tag{73}$$

Taking advantage of the following relationship

$$\frac{d}{dx} \frac{\delta f[q(x); q_d]}{\delta q(x)} = f^{(2)}[q(x); q_d]\dot{q}(x) \tag{74}$$

the second derivative of the free energy turns out to be

$$f^{(2)}[q(x); q_d]\dot{q}(x) = \frac{1}{2} \frac{d}{dx} \int dh \, P(h, x)f'(x, h)^2 \ . \tag{75}$$

Then, according to this second relation:

$$\frac{d}{dx} f^{(n)}[q(x); q_d] = f^{(n+1)}[q(x); q_d]\dot{q}(x) \qquad \text{for} \quad n \geq 2 \ , \tag{76}$$

the r.h.s of Eq. (75) can be re-expressed as

$$\frac{d}{dx} \int dh P(h, x)f'(x, h)^2 = \dot{q}(x) \int dh P(h, x)f''(x, h)^2 \ . \tag{77}$$

Gathering the two expressions together, one obtains

$$f^{(2)}[q(x); q_d]\dot{q}(x) = \frac{\dot{q}(x)}{2} \int dh P(h, x)f''(x, h)^2 \tag{78}$$

Therefore, if there exists an interval $\in [0, 1]$ above which $\dot{q}(x) \neq 0$, one can argue that the variation of the free energy function to the second order is

$$f^{(2)}[q(x); q_d] = \frac{1}{2} \int dh P(h, x)f''(x, h)^2 \ , \tag{79}$$

which is essentially equivalent to determining the vanishing value of the replicon mode of the stability matrix, *i.e.* the marginal stability condition for the RS solution. Hence, the breaking point can be simply obtained by the differentiation of the above equation with respect to $x$:

$$\frac{d}{dx} \int dh P(h, x)f''(x, h)^2 = \dot{q}(x)\left[dh P(h, x)f'''(x, h)^2 - 2x \int dh P(h, x)f''(x, h)^3\right] \tag{80}$$

Using again Eqs. (74)-(75), we impose $f^{(3)}[q(x); q_d] = 0$, from which we immediately get

$$\int dh P(h, x)f'''(x, h)^2 - 2x \int dh P(h, x)(f''(x, h))^3 = 0 \ , \tag{81}$$

hence in terms of the breaking point $x^*$:

$$x^* = \frac{dh P(h, x)f'''(x, h)^2}{2 \int dh P(, hx)f''(x, h)^3} \ . \tag{82}$$

# B Alternative computation of the breaking point by a third-order perturbative expansion

To study stability properties, we can develop the matrix $Q_{ab}$ and the external field $h_a$ around the symmetric solution namely with $Q_{ab} = q_d \delta_{ab} + q_0$ and $h_a = h$. Note that we have called $q$ the diagonal value and $t$ the off-diagonal contribution.

$$
\begin{aligned}
Q_{ab} &= (q_d + \rho_a)\delta_{ab} + q_0 + \sigma_{ab} \ , \\
h_a &= h + l_a
\end{aligned}
\tag{83}
$$

where $\sigma_{aa} = 0$ and $\sigma_{ab} = \sigma_{ba}$. Expanding the free energy according to the definition in Eq. (83), we obtain to the third order:

$$
\begin{aligned}
f^{(3)} = &-\frac{1}{6}\langle N_a N_b N_c \rangle_c \sum_{abc} l_a l_b l_c + \frac{\beta}{2}\langle N_a N_b N_c^2 \rangle_c \sum_{abc} l_a l_b \rho_c - \frac{\beta^2}{2}\langle N_a N_b^2 N_c^2 \rangle_c \sum_{abc} l_a \rho_b \rho_c + \\
&+ \frac{\beta^3}{6}\langle N_a^2 N_b^2 N_c^2 \rangle_c \sum_{abc} \rho_a \rho_b \rho_c + \frac{\beta}{2}\langle N_a N_b (N_c N_d) \rangle_c \sum_{ab,c\neq d} l_a l_b \sigma_{cd} + \\
&- \beta^2 \langle N_a N_b^2 (N_c N_d) \rangle_c \sum_{ab,c\neq d} l_a \rho_b \sigma_{cd} + \frac{\beta^3}{2}\langle N_a^2 N_b^2 (N_c N_d) \rangle_c \sum_{ab,c\neq d} \rho_a \rho_b \sigma_{cd} + \\
&- \frac{\beta^2}{2}\langle N_a (N_b N_c)(N_d N_e) \rangle_c \sum_{ab\neq c,d\neq e} l_a \rho_{bc} \sigma_{de} + \frac{\beta^3}{2}\langle N_a^2 (N_b N_c)(N_d N_e) \rangle_c \sum_{ab\neq c,d\neq e} \rho_a \sigma_{bc} \sigma_{de} + \\
&+ \frac{\beta^3}{6}\langle (N_a N_b)(N_c N_d)(N_e N_f) \rangle_c \sum_{a\neq b,c\neq d,e\neq f} \sigma_{ab} \sigma_{cd} \sigma_{ef} \ .
\end{aligned}
\tag{84}
$$

where three-point connected correlation functions can be simplied according to

$$
\langle ABC \rangle_c \equiv \langle ABC \rangle - \langle A \rangle \langle BC \rangle - \langle B \rangle \langle AC \rangle - \langle C \rangle \langle AB \rangle + 2\langle A \rangle \langle B \rangle \langle C \rangle \ .
\tag{85}
$$

In the following we resume a detailed computation that was similarly done in [39] in the case of the Random Replicant Model describing the evolution of an ensemble of replicants that evolve according to random interactions. Using Eq. (85) the third-order contribution of the free energy can be simplified as a combination of single averages

$$
\begin{aligned}
f^{(3)} = &-\frac{l^3}{6}\left[ \overline{\langle N^3 \rangle} - 3\overline{\langle N^2 \rangle \langle N \rangle} + 2\overline{\langle N \rangle^3} \right] + \frac{\beta l^2 \rho}{2}\left[ \overline{\langle N^4 \rangle} - 2\overline{\langle N^3 \rangle \langle N \rangle} - \overline{\langle N^2 \rangle^2} + \overline{\langle N \rangle^2 \langle N^2 \rangle} \right] + \\
&- \frac{\beta^2 l \rho^2}{2}\left[ \overline{\langle N^5 \rangle} - \overline{\langle N^4 \rangle \langle N \rangle} - 2\overline{\langle N^3 \rangle \langle N^2 \rangle} + 2\overline{\langle N^2 \rangle^2 \langle N \rangle} \right] + \\
&+ \frac{\beta^3 \rho^3}{6}\left[ \overline{\langle N^6 \rangle} - 3\overline{\langle N^4 \rangle \langle N^2 \rangle} + 2\overline{\langle N^2 \rangle^3} \right] + \\
&+ \frac{\beta l^2}{2}\left[ 6\overline{\langle N^3 \rangle \langle N \rangle} - 2\overline{\langle N^2 \rangle^2} - 10\overline{\langle N \rangle^2 \langle N^2 \rangle} + 6\overline{\langle N \rangle^4} + 4(\overline{\langle N^2 \rangle}^2 - \overline{\langle N \rangle^2} \cdot \overline{\langle N^2 \rangle}) \right] \sum_{\neq} \sigma_{ab} +
\end{aligned}
\tag{86}
$$

$$- \beta^2 l \rho \left[ 6\overline{\langle N^4\rangle\langle N\rangle} - 2\overline{\langle N^3\rangle\langle N^2\rangle} - 6\overline{\langle N\rangle^2\langle N^3\rangle} - 4\overline{\langle N^2\rangle^2\langle N\rangle} + 6\overline{\langle N\rangle^3\langle N^2\rangle} + \right.$$

$$\left. + 4\left( \overline{\langle N^2\rangle\langle N\rangle \cdot \langle N\rangle^2} - \overline{\langle N^3\rangle \cdot \langle N\rangle^2} \right) \right] \sum_{\neq} \sigma_{ab} +$$

$$+ \frac{\beta^3 \rho^2}{2} \left[ 6\overline{\langle N^5\rangle\langle N\rangle} - 2\overline{\langle N\rangle^2\langle N^4\rangle} - 2\overline{\langle N^3\rangle^2} - 8\overline{\langle N^3\rangle\langle N^2\rangle\langle N\rangle} + 6\overline{\langle N\rangle^2\langle N^2\rangle^2} + \right.$$

$$\left. + 4(\overline{\langle N^2\rangle^2 \cdot \langle N\rangle^2} - \overline{\langle N^4\rangle\langle N\rangle^2}) \right] \sum_{\neq} \sigma_{ab} +$$

$$- 2\beta^2 l \left[ \left( \overline{\langle N^3\rangle\langle N^2\rangle} - \overline{\langle N^2\rangle^2\langle N\rangle} + 2\overline{\langle N\rangle^3 \cdot \langle N\rangle^2} - 2\overline{\langle N^2\rangle\langle N\rangle \cdot \langle N\rangle^2} \right) \sum_{\neq} \sigma_{ab}^2 + \right.$$

$$+ \left( \overline{\langle N^2\rangle\langle N^3\rangle} + 2\overline{\langle N^2\rangle^2\langle N\rangle} - 3\overline{\langle N\rangle^3\langle N^2\rangle} + 4\overline{\langle N\rangle^3 \cdot \langle N\rangle^2} - 4\overline{\langle N^2\rangle\langle N\rangle \cdot \langle N\rangle^2} \right) \sum_{\neq} \sigma_{ab}\sigma_{bc} +$$

$$\left. + \left( \overline{\langle N\rangle^3\langle N^2\rangle} - \overline{\langle N\rangle^5} + \overline{\langle N\rangle^3 \cdot \langle N\rangle^2} - \overline{\langle N^2\rangle\langle N\rangle \cdot \langle N\rangle^2} \right) \sum_{\neq} \sigma_{ab}\sigma_{cd} \right] +$$

$$+ 2\beta^3 \rho \left[ \left( \overline{\langle N^4\rangle\langle N^2\rangle} - \overline{\langle N^2\rangle^3} + 2\overline{\langle N\rangle^2\langle N^2\rangle \cdot \langle N\rangle^2} - 2\overline{\langle N^3\rangle\langle N\rangle \cdot \langle N\rangle^2} \right) \sum_{\neq} \sigma_{ab}^2 + \right.$$

$$+ \left( \overline{\langle N\rangle^2\langle N^4\rangle} + 2\overline{\langle N^3\rangle\langle N^2\rangle\langle N\rangle} - 3\overline{\langle N\rangle^2\langle N^2\rangle^2} + 4\overline{\langle N\rangle^2\langle N^2\rangle \cdot \langle N\rangle^2} - 4\overline{\langle N^3\rangle\langle N\rangle \cdot \langle N\rangle^2} \right)$$

$$\left. \cdot \sum_{\neq} \sigma_{ab}\sigma_{bc} + \left( \overline{\langle N\rangle^3\langle N^3\rangle} - \overline{\langle N\rangle^4\langle N^2\rangle} + \overline{\langle N\rangle^2\langle N^2\rangle \cdot \langle N\rangle^2} - \overline{\langle N^3\rangle\langle N\rangle\langle N\rangle^2} \right) \sum_{\neq} \sigma_{ab}\sigma_{cd} \right] +$$

$$+ \frac{\beta^3}{6} \left[ 4\left( \overline{\langle N^3\rangle^2} - 3\overline{\langle N^2\rangle^2 \cdot \langle N\rangle^2} + 2\overline{\langle N\rangle^2}^3 \right) \sum_{\neq} \sigma_{ab}^3 + \right.$$

$$+ 24\left( \overline{\langle N^3\rangle\langle N^2\rangle\langle N\rangle} - 2\overline{\langle N\rangle^2\langle N^2\rangle \cdot \langle N\rangle^2} - \overline{\langle N^2\rangle^2 \cdot \langle N\rangle^2} + 2\overline{\langle N\rangle^2}^3 \right) \sum_{\neq} \sigma_{ab}^2\sigma_{bc} +$$

$$+ 8\left( \overline{\langle N^2\rangle^3} - 3\overline{\langle N\rangle^2\langle N^2\rangle \cdot \langle N\rangle^2} + 2\overline{\langle N\rangle^2}^3 \right) \sum_{\neq} \sigma_{ab}\sigma_{bc}\sigma_{ca} +$$

$$+ 6\left( \overline{\langle N^2\rangle^2\langle N\rangle^2} - \overline{\langle N^2\rangle^2 \cdot \langle N\rangle^2} - 2\overline{\langle N\rangle^4 \cdot \langle N\rangle^2} + 2\overline{\langle N\rangle^2}^3 \right) \sum_{\neq} \sigma_{ab}^2\sigma_{cd} +$$

$$+ 24\left( \overline{\langle N^2\rangle^2\langle N\rangle^2} - 2\overline{\langle N\rangle^2\langle N^2\rangle \cdot \langle N\rangle^2} - \overline{\langle N\rangle^4 \cdot \langle N\rangle^2} + 2\overline{\langle N\rangle^2}^3 \right) \sum_{\neq} \sigma_{ab}\sigma_{ac}\sigma_{bd} +$$

$$8\left( \overline{\langle N\rangle^3\langle N^3\rangle} - 3\overline{\langle N\rangle^2\langle N^2\rangle \cdot \langle N\rangle^2} + 2\overline{\langle N\rangle^2}^3 \right) \sum_{\neq} \sigma_{ab}\sigma_{ac}\sigma_{ad} +$$

$$+ 12\left( \overline{\langle N\rangle^4\langle N^2\rangle} - \overline{\langle N\rangle^2\langle N^2\rangle \cdot \langle N\rangle^2} - 2\overline{\langle N\rangle^4 \cdot \langle N\rangle^2} + 2\overline{\langle N\rangle^2}^3 \right) \sum_{\neq} \sigma_{ab}\sigma_{bc}\sigma_{de} +$$

$$\left. + \left( \overline{\langle N\rangle^6} - 3\overline{\langle N\rangle^4 \cdot \langle N\rangle^2} + 2\overline{\langle N\rangle^2}^3 \right) \sum_{\neq} \sigma_{ab}\sigma_{cd}\sigma_{ef} \right] \cdot$$

$$(87)$$

As the next step, the sum over the $\sigma$'s must be expressed in terms of the continuous function $q(x)$. One typically uses the hypothesis that such $\sigma$ matrices break the replica symmetry in a hierarchical way and then looks for the solution $q(x)$ that minimizes the free energy functional, written as a combination of second-order and third-order contributions as in Eq. (87). Therefore, by differentiating $\frac{\delta F[q]}{\partial q(x)} = 0$ with respect to $x$, one finds precisely Eq. (54) of the main text, provided $\dot{q}(x) \neq 0$.

## C Expansion in the zero demographic noise limit and marginal stability

In the following, we shall derive the marginal stability condition and its behavior in the zero demographic noise limit. The starting point is the analysis of the stability matrix, namely the second derivatives of the free energy with respect to the overlap matrix $Q_{ab}$:

$$-\frac{\partial^2 A}{\partial Q_{ab} \partial Q_{cd}} = \beta^2 \rho^2 \sigma^2 \delta_{(ab),(cd)} - (\beta^2 \rho^2 \sigma^2)^2 \overline{\langle N^a N^b, N^c N^d \rangle_c} \tag{88}$$

from which, in the RS Ansatz according to [49–51], the replicon eigenvalue reads

$$\begin{aligned}
\lambda_{\text{repl}} &= M_{ab,ab} - 2M_{ab,ac} + M_{ab,cd} = \\
&= (\beta\rho\sigma)^2 \left\{ 1 - (\beta\rho\sigma)^2 \overline{[\langle (N^a)^2 (N^b)^2 \rangle - 2\langle (N^a)^2 N^b N^c \rangle + \langle N^a N^b N^c N^d \rangle]} \right\}.
\end{aligned} \tag{89}$$

To prove the instability of the RS solution it is sufficient to show that the second term in the above expression is either greater than one or divergent under some conditions. The above expression can be rewritten in a more straightforward way by noticing that the three correlators correspond to the second and higher-order moment of the abundances averaged over the conditioned probability distribution. We focus then on the conditioned probability of the four-replica-index correlator, conditioned to the Gaussian variable $z$.

$$\langle (N^a)^2 (N^b)^2 \rangle_z = \frac{1}{Z(z)^4} \left( \int dN e^{-\beta \left[ -\left( \frac{\rho^2 \sigma^2}{2} \beta(q_d - q_0) + (r + m/K) \right) N^2 + \left( \rho\mu h + mr - z\rho\sigma\sqrt{q_0} \right) N + \rho N^3 \right]} N^2 \right)^2 \tag{90}$$

$$\langle N^a N^b N^c N^d \rangle_z = \frac{1}{Z(z)^4} \left( \int dN e^{-\beta \left[ -\left( \frac{\rho^2 \sigma^2}{2} \beta(q_d - q_0) + (r + m/K) \right) N^2 + \left( \rho\mu h + mr - z\rho\sigma\sqrt{q_0} \right) N + \rho N^3 \right]} N \right)^4 \tag{91}$$

$$\begin{aligned}
\langle (N^a)^2 N^b N^c \rangle_z = \frac{1}{Z(z)^4} &\left( \int dN e^{-\beta \left[ -\left( \frac{\rho^2 \sigma^2}{2} \beta(q_d - q_0) + (r + m/K) \right) N^2 + \left( \rho\mu h + mr - z\rho\sigma\sqrt{q_0} \right) N + \rho N^3 \right]} N^2 \right) \cdot \\
&\cdot \left( \int dN e^{-\beta \left[ -\left( \frac{\rho^2 \sigma^2}{2} \beta(q_d - q_0) + (r + m/K) \right) N^2 + \left( \rho\mu h + mr - z\rho\sigma\sqrt{q_0} \right) N + \rho N^3 \right]} N \right)^2
\end{aligned} \tag{92}$$

The last correlator $\langle (N^a)^2 N^b N^c \rangle_z$ is a combination of the previous ones: its expression is slightly more involved but still exactly calculable. For the sake of compactness, in all the expressions above we have indicated the partition function as

$$Z(z) = \int dN e^{-\beta \left[ -\left( \frac{\rho^2 \sigma^2}{2} \beta(q_d - q_0) + (r + m/K) \right) N^2 + \left( \rho\mu h + mr - z\rho\sigma\sqrt{q_0} \right) N + \rho N^3 \right]}. \tag{93}$$

Under the hypothesis that the partition function can be reasonably well approximated by a Gaussian integral, the denominator turns out to be an error function modulated by an exponential factor. Indeed, in the $\beta \to \infty$ limit, we can safely expand the term at the exponent around the saddle-point value $N^*$, which corresponds to a harmonic approximation in $(N - N^*)$, that is:

$$
\begin{aligned}
(b - zc)N - aN^2 + \rho N^3|_N^* \approx\ & N^* \left( b - aN^* - cz + (N^*)^2\rho \right) + \\
& + \left( b - 2aN^* - cz + 3(N^*)^2\rho \right)(N - N^*) \\
& + (-a + 3N^*\rho)(N - N^*)^2 + O(N - N^*)^3\ .
\end{aligned}
\tag{94}
$$

Therefore, the normalization can be written as

$$
Z(z) = \int dN \theta(N) e^{-\beta\left[N^*\left(b-aN^*-cz+(N^*)^2\rho\right)+\left(b-2aN^*-cz+3(N^*)^2\rho\right)(N-N^*)+(-a+3N^*\rho)(N-N^*)^2\right]}
\tag{95}
$$

which is exactly calculable because of the Gaussian form of the integral. Then, the average species abundance reads

$$
A(z) = \int_0^\infty dN e^{-\beta\left[N^*\left(b-aN^*-cz+(N^*)^2\rho\right)+\left(b-2aN^*-cz+3(N^*)^2\rho\right)(N-N^*)+(-a+3N^*\rho)(N-N^*)^2\right]} N
$$

$$
= \frac{e^{-\beta \frac{(b-cz)^2+2(N^*)^2(-3b+2aN^*+3cz)\rho-3(N^*)^4\rho^2}{4(a-3N^*\rho)}}}{4[-\beta(a-3N^*\rho)]^{3/2}} \Bigg\{ 2e^{\beta\frac{(b-cz-3(N^*)^2\rho)^2}{4(a-3N^*\rho)}} \sqrt{-\beta(a-3N^*\rho)} +
$$

$$
\sqrt{\pi}\beta(-b + cz + 3(N^*)^2\rho) \left[ 1 + \mathrm{Erf}\left( \frac{\beta(-b+cz+3(N^*)^2\rho)}{2\sqrt{-\beta(a-3N^*\rho)}} \right) \right] \Bigg\}
\tag{96}
$$

and its second-order moment respectively:

$$
B(z) = \int_0^\infty dN e^{-\beta\left[N^*\left(b-aN^*-cz+(N^*)^2\rho\right)+\left(b-2aN^*-cz+3(N^*)^2\rho\right)(N-N^*)+(-a+3N^*\rho)(N-N^*)^2\right]} N^2
$$

$$
= -\frac{e^{-\beta\rho(N^*)^3}}{8[-\beta(a-3N^*\rho)]^{5/2}}\beta\Bigg\{ 2\sqrt{-\beta(a-3N^*\rho)}\left(b - cz - 3(N^*)^2\rho\right) +
$$

$$
+ e^{-\frac{\beta(b-cz-3(N^*)^2\rho)^2}{4(a-3N^*\rho)}}\sqrt{\pi}\left[ 2a - 6N^*\rho - \beta(b - cz - 3(N^*)^2\rho)^2 \right] \cdot
$$

$$
\cdot \mathrm{Erfc}\left( \frac{\beta(b-cz-3(N^*)^2\rho)}{2\sqrt{-\beta(a-3N^*\rho)}} \right) \Bigg\}
\tag{97}
$$

Then, the last correlator can be expressed as a combination of the first two pieces:

$$
C(z) = B(z) \cdot A(z)^2\ .
\tag{98}
$$

Gathering all the information on the three correlators together, we end up with the expression of the replicon to be averaged over the Gaussian variable $z$

$$
\left[ \langle (N^a)^2(N^b)^2 \rangle - 2\langle (N^a)^2 N^b N^c \rangle + \langle N^a N^b N^c N^d \rangle \right] = \int \mathcal{D}z \frac{B(z)^2 - 2C(z) + A(z)^4}{Z^4}
\tag{99}
$$

Despite the fact that the resulting expression might be rather involved, we can quite easily shown that it tends to diverge irrespective of the parameters $a$, $b$, $c$, $\rho$, as $\beta \gg 1$.

We aim now at generalizing the aforementioned computation as a function of a generic potential $V_i(N_i)$, beyond the simple Lotka-Volterra and cubic interaction cases. We thus express the starting Hamiltonian in the RS Ansatz as:

$$H_{\mathrm{RS}}(N_i, z_i) = V_i(N_i) - \frac{\rho^2 \sigma^2 \beta (q_d - q_0)}{2} N_i^2 + (\rho \mu h - z_i \rho \sigma \sqrt{q_0}) N_i \tag{100}$$

where for the sake of simplicity we can absorb both contributions of the linear shift in $N_i$ in a single term $\tilde{z}_i c$. By employing a harmonic approximation in $(N - N^*)$, we eventually end up with:

$$\tilde{H}_{\mathrm{RS}}(N) = V(N) - \frac{\rho^2 \sigma^2 \Delta q}{2} N^2 + \tilde{z} c N \approx \left[ c N^* \tilde{z} - \frac{1}{2} \Delta q (N^*)^2 \rho^2 \sigma^2 + V(N^*) \right] +$$
$$+ \left[ c \tilde{z} - \Delta q N^* \rho^2 \sigma^2 + V'(N^*) \right] (N - N^*) + + \frac{1}{2} \left[ -\Delta q \rho^2 \sigma^2 + V''(N^*) \right] (N - N^*)^2 . \tag{101}$$

The corresponding partition function becomes then

$$Z = \int dN \theta(N) e^{-\beta \tilde{H}_{\mathrm{RS}}} = \frac{1}{\sqrt{\beta \left( -\Delta q \rho^2 \sigma^2 + V''(N^*) \right)}} \cdot$$
$$\cdot e^{-\beta \frac{\left[ c^2 \tilde{z}^2 + V'(N^*) \left( 2c\tilde{z} - 2\Delta q N^* \rho^2 \sigma^2 + V'(N^*) \right) + 2V(N^*) \left( \Delta q \rho^2 \sigma^2 - V''(N^*) \right) + N^* \left( -2c\tilde{z} + \Delta q N^* \rho^2 \sigma^2 \right) V''(N^*) \right]}{2 \left( \Delta q \rho^2 \sigma^2 - V''(N^*) \right)}} \cdot$$
$$\cdot \sqrt{\pi/2} \, \mathrm{Erfc} \left( \frac{\beta \left( c\tilde{z} + V'(N^*) - N^* V''(N^*) \right)}{\sqrt{2 \left( \beta(-\Delta q \rho^2 \sigma^2 + V''(N^*)) \right)}} \right) \tag{102}$$

Considering now the asymptotic expansion of the complementary error function in the $\beta \to \infty$ limit, which can be simplified with the other exponential factor, we can claim that the only relevant term is the normalization $1/\sqrt{\beta \left( -\Delta q \rho^2 \sigma^2 + V''(N^*) \right)}$. Note again that the factor $\Delta q = \beta(q_d - q_0)$ has been introduced to properly deal with $O(1)$ quantities in this regime.

We focus then on the other terms, which are expressed as averages over the auxiliary Gaussian variable of the conditioned probability of the four-replica-index correlators, then conditioned to $z$. Then, neglecting exponentially small factors, we can rewrite the resulting combination, as shown in Eq. (99), in the following way:

$$(\rho \sigma)^2 \overline{\left[ \langle (N^a)^2 (N^b)^2 \rangle - 2 \langle (N^a)^2 N^b N^c \rangle + \langle N^a N^b N^c N^d \rangle \right]} = \overline{\frac{\rho^2 \sigma^2}{\left( -\Delta q \rho^2 \sigma^2 + V''(N^*) \right)^2}} . \tag{103}$$

where the overline denotes the average over the species distribution $P(N^*)$. Note that, in the $\beta \to \infty$ limit, most of the terms cancel out with each other because the error functions themselves tend to one in this regime.

In the simplest Lotka-Volterra logistic growth case one immediately recognizes that this equation implies

$$\frac{\rho^2 \sigma^2}{\rho^2 \left( 1 - \sigma^2 \Delta q \right)^2} = \frac{\sigma^2}{\left( 1 - \sigma^2 \Delta q \right)^2} \tag{104}$$

because of $\left(-\Delta q\rho\sigma^2 + V''(N^*)\right) = H''(N^*)$, in perfect agreement with the results obtained in [7].

# D    Coupled potentials: argument on the Bethe lattice

To show that only diagonal terms matter in the resulting formulation, we can for instance rephrase the model proposed in Eq. (70) on a Bethe lattice, namely on a random graph with a locally tree-like structure. Eq. (70) should be then generalized for all site on the lattice.

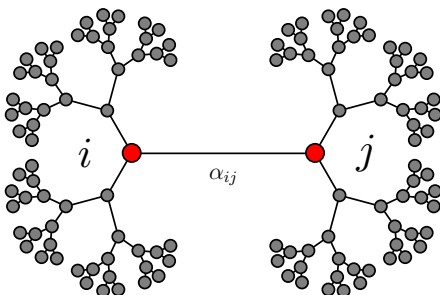

Figure 7: Part of the Bethe lattice connecting sites $i$ and $j$ with coupling $\alpha_{ij}$. The lattice is assumed to be ideally infinite with all sites equivalent.

Compared to fully connected (FC) models, which are typically simpler to solve, the Bethe lattice has the advantage of being more akin to realistic finite-dimensional models because of its finite connectivity. On the other hand, from the point of view of critical phenomena, a phase transition on the Bethe lattice is mean-field-like: hence, if such a transition does also exist in the large-connectivity/FC limit, then the critical exponents should be the same as those of the corresponding continuum field theory. Generically, the density of eigenvalues $\lambda_\gamma$ can be obtained through

$$\rho(E) = \frac{1}{S} \sum_\gamma \delta(E - \lambda_\gamma) \tag{105}$$

where $S$ denotes the total number of species, $i.e.$ of sites in the current notation. Its expression is strictly related to the resolvent matrix $G(E)$

$$\rho(E) = \lim_{\epsilon \to 0} \frac{1}{\pi S} \Im \operatorname{Tr} G(E + i\epsilon) \tag{106}$$

where $\Im(\cdot)$ is the imaginary part. To define the spectral properties of the Hamiltonian operator we would need to better investigate the resolvent matrix structure, which is defined as

$$G_{ij}(z) = \left(\frac{1}{H - z\mathbb{1}}\right). \tag{107}$$

Its diagonal part turns out to be

$$G_{ii}(z) = \frac{1}{H_{ii} - z - \sum_{k=1}^{K} \sum_{j=1}^{K} \alpha_{kj}^2 G_{k,j}(z)} \;, \tag{108}$$

as it can be proven by means of a perturbative expansion, in the same spirit of the Dyson equation in diagrammatic theory, or by Random Matrix Theory (RMT) properties, such as the *Schur's complement formula* on inverse matrices [52]. By perturbation theory, one basically notices that the two contributions at the denominator can be split into a diagonal part plus a second one written a sum running over all paths in the lattice that connects $i$ and $j$, for which site $i$ is excluded. In this light, the last contribution of Eq. (108) can be further simplified leading to the following expression for the diagonal part of the resolvent

$$G_{ii}(z) = \frac{1}{H_{ii} - z - \sum\limits_{k=1}^{K} G_{k,k}(z)} \ ,$$ (109)

where $G_{k,k}(z)$ stands for the cavity correlator (or Green function) with $\alpha_{kk} = 1$. Because in the large-$S$ limit off-diagonal terms(with $k \neq j$) are negligible, we can focus only on diagonal terms. More precisely, on every graph with a tree-like topology, sites $k$ and $j$ turn out to be directly connected once that $i$ is removed.

For the sake of brevity, we will denote the diagonal element of the resolvent matrix as $G(z)$, we can also deduce the spectral properties through the computation of the density $\rho(E)$, see Eq. (106). Hence, the introduction of an infinitesimal field on each $N_i$ will result in a term $\delta N_i$, which can be dealt with using the same strategy as in Sec. (5.2). The energy variation will be then

$$\delta \mathcal{E} = \frac{1}{2} \sum_i \delta N_i^2 V_{\text{eff}}''(N_i^*)$$ (110)

where, again, we can consider only diagonal terms of the interaction matrix in Eq. (71). The argument follows precisely as in the main text giving a criterion for marginal stability in terms of the distribution $P(V_{\text{eff}}''(N^*))$.

## E  Connections with a disordered instance of two-level system (TLS)

In a recent paper [53], a generalization of the Kühn and Horstmann (KH) model combined with the Gurevich, Parshin and Schober (GPS) three-dimensional lattice version in the presence of random interactions and a constant external field has been proposed in order to study zero-temperature vibrational modes of a specifically designed disordered system. The model is formulated as a collection of anharmonic oscillators subject to a random distribution $p(\kappa)$ for the stiffnesses $\kappa$, which are taken uniform in the interval $[\kappa_{\min}, \kappa_{\max}]$ for positive or zero values. In this formulation, the Hamiltonian reads

$$H_{\text{KHGPS}} = \sum_{i<j} J_{ij} x_i x_j + \frac{1}{2} \sum_i \kappa_i x_i^2 + \frac{1}{4!} \sum_i x_i^4 - h \sum_i x_i$$ (111)

where the couplings $J_{ij}$ are i.i.d variables, Gaussian distributed with zero mean and variance $J^2/N$. If the lower edge of the stiffness support is strictly positive, the spectral density is gapped at small coupling strength leading to a purely convex landscape. Conversely, upon increasing the coupling strength, a phase transition takes place according to which the lower edge of the spectral density touches zero as a signature of marginality.

Specifically in the $T \to 0$ limit, one can define an effective potential $v_{\text{eff}}(x) \equiv \frac{x^4}{4!} + \frac{m}{2}x^2 - (f+h)x$, where $f$ is a random force while $m = \kappa - J^2\chi$, where $\chi$ is related to the correlation functions between different replicas and should be determined self-consistently.

In [53] it has been shown that at small external field the density of states is purely quadratic $D(\omega) \sim \omega^2$ as correctly predicted by mean-field models, and the corresponding modes are delocalized. On the other hand, upon increasing the external field, a quartic pseudogap $D(\omega) \sim \omega^4$ takes place associated with the appearance of localized modes. The two regimes are separated by a special point on the critical line in the two-dimensional plane for the applied magnetic field versus the interaction strength $(J, h)$. The strength $J$ plays a crucial role because for sufficiently small couplings the spectrum remains gapped with a twofold scenario for the density of states (either quartic dependence in $\omega$ or standard quadratic trend as observed in most mean-field spin-glass systems). Moreover, if the minimal stiffness – defining the lower edge of the support of $p(\kappa)$ – shrinks to zero, the critical transition line $J_c(h)$ shifts to the left as well allowing only for a RSB phase, irrespective of the strength $J$.

# F  Toy model of coupled oscillators: a field theory approach

To highlight the connection with our model in presence of a cubic potential, we will present an argument inspired by a system of coupled oscillators subject to a given interacting potential. We shall then generalize this picture considering an additional quartic-order contribution which accounts specifically for a double-well potential.

Therefore, we can map the parameters of the KHGPS model – notably the marginal stability condition for the stiffness – into a weak or strong Allee effect depending on the selected value of the threshold $m$ and the generalized field. The interplay between the parameter $m$ and the field is indeed responsible for a different phenomenology associated with a single or a double-well potential in the species abundances, respectively.

To give a more concrete example, we first consider a system on a linear chain defined by $N$ identical massive particles (with given mass $m$) oscillating around their equilibrium positions and subject to a harmonic potential. The Hamiltonian can be written then as

$$\mathcal{H} = \sum_{n=1}^{N} \frac{p_n^2}{2m} + \frac{m\omega^2}{2}(q_n - q_{n-1})^2 + \frac{m\omega_0^2}{2}q_n^2 \tag{112}$$

where the second term represents the nearest-neighbor coupling, whereas the last term simply reproduces the Hooke contribution. According to quantum mechanics formalism, we can resort to the usual relations for the evolution of positions and momenta:

$$\begin{cases} \dot{q}_n(t) = \frac{p_n(t)}{m} \\ \dot{p}_n(t) = m\omega^2 \left[ q_{n+1}(t) + q_{n-1}(t) - 2q_n(t) \right] - m\omega_0^2 q_n(t) \end{cases} \tag{113}$$

In the continuum limit, particles are function of both position and time, therefore by the introduction of a field:

$$\ddot{q}(x,t) = \omega^2 a^2 \left[ \frac{q(x+a,t) - q(x,t) - (q(x,t) - q(x-a,t))}{a^2} \right] - \omega_0^2 q(x,t) \tag{114}$$

which in the limit $a \to 0$, $N \to \infty$ implies:

$$\lim_{a \to 0, N \to \infty} \ddot{q}(x,t) \Rightarrow \quad \ddot{q}(x,t) - \omega^2 a^2 \frac{\partial^2}{\partial x^2} q(x,t) + \omega_0^2 q(x,t) = 0 \tag{115}$$

If we define $\omega a \equiv v$ and set $\omega_0 = 0$, we can exactly recover the wave equation in one dimension with $\ddot{q}(x, t) - v^2 \partial^2/\partial x^2 q(x, t) = 0$. For the sake of convenience we nevertheless choose the rescaling $(\omega/v)^2 = m^2$, $q(x, t) = \phi(x)$ and $v = c$, which allows us to end up with a Klein-Gordon equation for bosonic fields. In the covariant formulation, the equation reads:

$$\partial_\mu \partial^\mu \phi(x) + m^2 \phi(x) = 0 . \tag{116}$$

from which one can derive the corresponding Klein-Gordon action in a few passages

$$S[\phi] = \int d^4x \; \left( -\frac{1}{2} \partial_\mu \partial^\mu \phi - \frac{m^2}{2} \phi\phi \right) \tag{117}$$

Note that the mass term $m^2$ plays a crucial role as strictly connected to the appearance of Goldstone modes, hence of a marginally stable solution. Coming back to the model defined in Eq. (111), this would lead to a condition for the stiffness term $k \leftrightarrow m^2$: when the support of $k$ touches zero, an intrinsic instability emerges as a consequence of a spontaneous symmetry breaking.

## F.1 Further analysis of a system of anharmonic oscillators subject to quartic-order perturbation

In the previous Section, we have briefly discussed how to recover the instability condition by the analysis of the stiffness distribution and the determination of the vanishing behavior of the lower edge of their support. In the following, we will show one possible strategy to go beyond a model of purely harmonic oscillators by means of the introduction of quartic-order contribution to be eventually treated in perturbation theory. The techniques we are going to use in the following are standard and based on the computation of the first corrections to the ground state energy in a quantum mechanical system of anharmonic oscillators. Following the same hypothesis of [54], in the $T \to 0$ limit we can write the solution as a problem of decoupled oscillators subject to a given effective potential.

$$w = \int \mathcal{D}q \; \exp\left\{ - \int d\tau \left( \frac{m\dot{q}^2}{2} + \frac{1}{2} m\omega^2 q^2 + \frac{\lambda}{4!} q^4 \right) \right\} \tag{118}$$

The following notation $w[J]^0$ stands for the generating function without the quartic term but in presence of an external force:

$$w^{(0)}[J] = \int \mathcal{D}q \; \exp\left\{ - \int d\tau \left( \frac{m\dot{q}^2}{2} + \frac{1}{2} m\omega^2 q^2 - Jq \right) \right\} \tag{119}$$

According to this reshuffling, we can obtain a compact expression as a function of cumulants of $(\delta/\delta J)^4$, based on the following identity:

$$e^{-\int d\tau \frac{\lambda}{4!} \left( \frac{\delta}{\delta J} \right)^4} w^{(0)}[J] = \int \mathcal{D}q \exp\left\{ - \int d\tau \frac{\lambda}{4!} q^4 \right\} e^{-S^{(0)}[J]} =$$
$$= \int \mathcal{D}q \exp\left\{ - \int d\tau \left( \frac{1}{2} m\dot{q}^2 + \frac{1}{2} m\omega^2 q^2 + \frac{\lambda}{4!} q^4 - Jq \right) \right\} \tag{120}$$

from which we can expand the exponential in power series and eventually take $J = 0$ to recover the initial expression. To compute Eq. (119), we consider the equation of motion in presence of the external source $J$ and suppose to be able to find the classical solution for $q$

$$m\ddot{q}(\tau) = m\omega^2 q(\tau) - J(\tau) \tag{121}$$

with $q = 0$ at the initial time. In a few passages, we can rewrite the expression for $w^{(0)}[J]$ as:

$$w^{(0)}[J] \propto A e^{-S[J]} \tag{122}$$

where

$$S[J] = \int d\tau \left( \frac{1}{2} m \dot{q}_{\text{clas}}^2 + \frac{1}{2} m \omega^2 q_{\text{class}}^2 - J q_{\text{class}} \right) \tag{123}$$

Then, we can simply integrate the first term by part, take advantage of the equation of motion and resort to the Green's function formalism to write the resulting solution. In this way, we end up with $m \left( \frac{d^2}{d\tau^2} - \omega^2 \right) G(\tau, \tau') = \delta(\tau - \tau')$ where $G(\tau, \tau')$ can be expressed in the Fourier space as $G(\tau, \tau') \equiv - \int_{-\infty}^{\infty} \frac{dk}{2\pi} \frac{1}{m(k^2+\omega^2)} e^{ik(\tau-\tau')}$. Accordingly, one can deduce the first-order correction to $w[J]$, hence the ground state energy from the evaluation of its logarithm. One can thus show that the first energetic contribution of a system of decoupled oscillators, $\mathcal{E}_0 = \frac{\hbar h \omega}{2}$, is actually corrected by a term that is inversely proportional to $m^2 \omega^2$ confirming the divergence of the asymptotic expansion as $m^2 \to 0$.

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
