# Peer review of "Effects of intraspecific cooperative interactions in large ecosystems"

_SciPost Physics_

## Round 2 · Referee Report · Anonymous (Referee 1) · 2021-7-18

Strengths

  • In-depth analysis of an interesting variation of the random Lotka Volterra model with random interaction matrix
  • New phase diagram identified
  • Very well written
  • Nice summary in Section 6.

Weaknesses

  • A little long, could move some of the technical analysis to an appendix so that the reader can access the main results more easily.
  • A little more could be said about the ecological relevance and consequences of the main findings.

Report

The authors present an equilibrium statistical physics analysis of an interesting extension of the Lotka-Volterra model with random interactions. The paper is (as far as I can tell from my reading) sound and the work is novel. It is also very well written.

As one minor comment one could say that the paper is very long, and that some of the technical analysis could perhaps be relegated to an appendix or supplement. This would make it easier for the reader to understand the main results (readers who are not expert in replica calculations would probably appreciate this). Having said this, the authors provide a nice summary of their findings in Section 6.

As a further comment, I am not entirely sure how applicable these results are in areas such as genetics, cancer evolution, epidemiology etc (all mentioned by the authors in the conclusions).

The authors could also say a little more about what their results mean for ecological communities. E.g. what should one look for in real ecosystems to perhaps confirm some of the results, e.g. how would one know what phase a real ecosystem is in?

But overall, the study is presented accurately as a theoretical statistical mechanics analysis of the model. The authors may optionally think about modifying such statements.

I am instructed to comment on the "Expectations" by the journal (at least one of them must be met).

  • "Detail a groundbreaking theoretical/experimental/computational discovery"

There are significant scientific advances here, but I do not know what "groundbreaking" means. I would need more information on how SciPost defines this.

  • "Present a breakthrough on a previously-identified and long-standing research stumbling block;"

Again, the journal would have to let me know what level of novelty is required to qualify as a "breakthrough". If coming up with an interesting and motivated variation of an existing model, addressing a difficult technical problem and identifying a new type of phase diagram qualifies then the criterion is met. If a discovery of the type "find the Higgs boson" is required to qualify as "breakthrough on a long-standing stumbling block", then I am afraid to say (and I hope the authors forgive me) that this paper is not quite at that level.

  • "Open a new pathway in an existing or a new research direction, with clear potential for multipronged follow-up work"

I think the paper significantly advances the field, but again the formulation of the criterion makes it hard for me to assess this. I do not know what "multipronged follow-up work" means. I would expect this work to trigger follow-up work in the area of random ecological communities as well as perhaps in spin glass or glass physics.

  • "Provide a novel and synergetic link between different research areas." The paper links different fields, namely theoretical ecology and spin glass physics. This link is "synergetic", but I am not sure it qualifies as "novel", as there is a body of work at this interface already.

The "General acceptance criteria" (which I understand are all required) are all fulfilled, e.g.

  • "Be written in a clear and intelligible way, free of unnecessary jargon, ambiguities and misrepresentations"
  • "Contain a detailed abstract and introduction explaining the context of the problem and objectively summarizing the achievements;"
  • "Provide sufficient details (inside the bulk sections or in appendices) so that arguments and derivations can be reproduced by qualified experts;" -"Provide citations to relevant literature in a way that is as representative and complete as possible"
  • "Provide (directly in appendices, or via links to external repositories) all reproducibility-enabling resources: explicit details of experimental protocols, datasets and processing methods, processed data and code snippets used to produce figures, etc.;"
  • "Contain a clear conclusion summarizing the results (with objective statements on their reach and limitations) and offering perspectives for future work."

Overall conclusion:

This is a very well-written, in-depth analysis of an interesting variant of the random Lotka-Volterra model. This will be of interest to a number of experts in random Lotka-Volterra models and to the spin glass and glass community. There may also be interest from theoretical ecologists, although these will probably find it hard to access and understand the main findings.

The paper therefore deserves to be published in a reputable outlet, essentially in its present form (perhaps after moving some of the calculations to an appendix or supplement).

Unfortunately the criteria SciPost presents me with are a little ambiguous, so I cannot tell if the expectations are met or not. If this had been submitted to Physical Review E for example, I would have said "publish as is" (with potential very minor corrections). If this had been submitted to say Physical Review X or PNAS, which claim to only publish truly exceptional papers, I would be more cautious. At the very least I would have asked the authors to explain the relevance of their work outside statistical physics in more detail, and in a form understandable to a wider readership. It really depends on what SciPost expects.

Requested changes

Potentially move some of the calculations to an appendix or supplement.

Potentially comment on ecological interpretation of the main findings, and how they could be tested in real ecosystems.

[Both of these are optional. If left in the current form the readership will probably be limited mostly to experts, so it is really a question as to how broad an audience the authors would like to reach.]

  • validity: high
  • significance: high
  • originality: high
  • clarity: good
  • formatting: good
  • grammar: excellent

Author:  Ada Altieri  on 2021-09-10  [id 1746]

(in reply to Report 1 on 2021-07-18)
Category:
answer to question

We strongly thank the referee for their extremely positive review and for highlighting multiple strengths, including an in-depth theoretical analysis for the Allee effect and completely new phase diagrams.
Although the referee stressed the optionality of proposed changes to avoid unnecessary loss of time, we decided to follow their suggestions as much as possible.
We have therefore opted, on the one hand, for more fluent and smooth reading of the paper thus moving some technical parts to the Appendix, and, on the other hand, for a better explanation of possible experimental outcomes to be related to our research.
Please find below a brief list of changes, which will be implemented in red in the resubmitted version of the manuscript.

i) We have substantially modified Sec. 3.1 by removing some technicalities and shifting all replica computations to Appendix A. Similarly, we have reduced Sec. 4.1 and left room for discussions about finite and diverging susceptibilities.
Furthermore, specific formulas concerning the breaking point computation in Sec. 4.2.1 have been better detailed in Appendix C.

ii) On the other hand, we have extended the conclusive section by discussing experimental implications of our results, notably: i) to infer real system parameters and investigate bi-stability criteria; ii) to concretely test the emergence of marginally stable equilibria at low population noise and high heterogeneity; iii) to analyze functional responses under local single-species perturbations.

We hope to have satisfactorily addressed all concerns and to be now on the optimal path for reaching a wider target.

---

## Round 2 · Referee Report · Anonymous (Referee 2) · 2021-8-30

Strengths

1- Interesting application of Fokker-Planck type equations 2-Good theoretical analysis

Weaknesses

1- Modeling assumptions are quite "ad hoc"

Report

I retain that the paper satisfies the requirements.

Requested changes

  • The choice of the potential would require better motivations for the ecological community

  • validity: high
  • significance: good
  • originality: high
  • clarity: high
  • formatting: excellent
  • grammar: good

Author:  Ada Altieri  on 2021-09-10  [id 1747]

(in reply to Report 2 on 2021-08-30)

We warmly thank the referee for accepting the manuscript in the current form and judging the format to be excellent for publication in SciPost.
Please find below a brief description of the implemented changes, in particular to emphasize the ecological relevance of a cubic potential in the species abundances with respect to a Lotka-Volterra-like logistic growth function.

i) After the definition of Eq. (2), we have better motivated our choice of the potential and its interest in the ecological realm. See from the text: "More concretely, the potential (2) allows us to capture the salient phenomena associated with the Allee effect. Populations with abundance greater than the limit population size will increase up to their carrying capacity, whereas those with a lower abundance will decline to extinction (see also Fig. 1). The carrying capacity turns out to be then a stable fixed point for both strong and weak Allee effects, and the logistic growth. Conversely, extinction is stable only for the strong Allee effect thus leading to a first difference both with respect to the weak variant and to a logistic behavior. This bistability -- as already mentioned, closely related to the mutualism of interactions -- is essentially determined by the initial condition. As a consequence, even a small perturbation in the initial population can be crucial and dramatically affect the behavior of ecosystems characterized by a strong Allee effect.”

ii) Moreover, a more in-depth discussion of well-controlled microbial community experiments has been added in the Conclusions. To give a specific example:
"Experimental studies on microbial populations [citations] can bring more power to theoretical predictions as well as provide a means to explore new ideas in evolutionary biology. One might consider a chemostat reactor in which changing the availability and concentration of nutrients (more nutrients would correspond to more heterogeneous interactions in our jargon) is responsible to trigger long-lasting modifications in the community composition. Upon increasing the number of available nutrients, one can extrapolate information on the mean abundance - averaged over all the samples - as a function of time as well as the correlation between abundances at two different times. From the analysis of dynamical correlation functions the emergence of a bunch of plateaus might be highlighted, corresponding to dynamical arrest at different timescales, hence to a multi-structured organization of the equilibria from a landscape perspective.
Furthermore, studies on the so-called alternative stable states and regime shifts are gaining an increasing popularity, especially in the context of the human gut microbiome [citations] as an effort to explain the enormous variability observed both within and among gut microbial communities."

---

## Editorial Decision

resubmitted